# Advancing Open-Set Domain Generalization Using Evidential Bi-Level Hardest Domain Scheduler

**Kunyu Peng**[1], **Di Wen**[1], **Kailun Yang**[2],[*] **Ao Luo**[3], **Yufan Chen**[1], **Jia Fu**[4,5],
**M. Saquib Sarfraz**[1,6], **Alina Roitberg**[7], **Rainer Stiefelhagen**[1]
[1]Karlsruhe Institute of Technology [2]Hunan University [3]Waseda University
[4]KTH Royal Institute of Technology [5]RISE Research Institutes of Sweden
[6]Mercedes-Benz Tech Innovation [7]University of Stuttgart

## Abstract

In Open-Set Domain Generalization (OSDG), the model is exposed to both new variations of data appearance (domains) and open-set conditions, where both known and novel categories are present at test time. The challenges of this task arise from the dual need to generalize across diverse domains and accurately quantify category novelty, which is critical for applications in dynamic environments. Recently, meta-learning techniques have demonstrated superior results in OSDG, effectively orchestrating the meta-train and -test tasks by employing varied random categories and predefined domain partition strategies. These approaches prioritize a well-designed training schedule over traditional methods that focus primarily on data augmentation and the enhancement of discriminative feature learning. The prevailing meta-learning models in OSDG typically utilize a predefined sequential domain scheduler to structure data partitions. However, a crucial aspect that remains inadequately explored is the influence brought by strategies of domain schedulers during training. In this paper, we observe that an adaptive domain scheduler benefits more in OSDG compared with prefixed sequential and random domain schedulers. We propose the **E**vidential **Bi-L**evel **Ha**rdest **D**omain **S**cheduler (EBiL-HaDS) to achieve an adaptive domain scheduler. This method strategically sequences domains by assessing their reliabilities in utilizing a follower network, trained with confidence scores learned in an evidential manner, regularized by max rebiasing discrepancy, and optimized in a bi-level manner. We verify our approach on three OSDG benchmarks, *i.e.*, PACS, DigitsDG, and OfficeHome. The results show that our method substantially improves OSDG performance and achieves more discriminative embeddings for both the seen and unseen categories, underscoring the advantage of a judicious domain scheduler for the generalizability to unseen domains and unseen categories. The source code is publicly available at https://github.com/KPeng9510/EBiL-HaDS.

## 1 Introduction

Open-Set Domain Generalization (OSDG) is a challenging task where the model is exposed to both: domain shift and category shift. Recent OSDG works often take a meta-learning approach [54, 46] which simulates different cross-domain learning tasks during training. These methods conventionally use a *predefined* sequential domain scheduler to create meta-train and meta-test domains within each minibatch. But is fixing the meta-learning domain schedule a priori the best way to go? As a step to explore this, our work investigates the new idea of *adaptive domain scheduler*, which dynamically adjusts the training order based on ongoing model performance and domain difficulty.

---

[*]Correspondence: kailun.yang@hnu.edu.cn

38th Conference on Neural Information Processing Systems (NeurIPS 2024).

OSDG is critical for many real-world applications with changing conditions, ranging from health-care [33] and security [5] to autonomous driving [19]. Despite the remarkable success of deep learning, the recognition quality often deteriorates when facing out-of-distribution samples. This problem is amplified in OSDG settings, where the model faces a dual challenge of identifying and rejecting unseen categories, *e.g.*, by delivering low confidence score in such cases [27, 46] while simultaneously generalizing well to unseen data appearances (domain shift). Historically, research efforts in OSDG have predominantly focused on the latter, developing methods to adapt models to varying domain conditions. Strategies to improve domain generalization include the use of Generative Adversarial Networks (GANs) [4], contrastive learning [65], and metric learning [27]. MLDG [46] for the first time proposed to use meta-learning to handle OSDG tasks. However, all these works follow the OSDG protocols where different source domains preserve different distributions of known categories, which diverges the domain gaps for different categories. This divergence makes the challenge more inclined towards the domain generalization aspect. Wang *et al.* [54] revised these benchmarks for OSDG, standardizing the category distribution across source domains to achieve a more balanced evaluation of both domain generalization and open set recognition challenges. This revision includes established open-set recognition methods such as Adversarial Reciprocal Points Learning (ARPL) [8]. Recently, Wang *et al.* [54] introduced an effective meta-learning approach named MEDIC, featuring a binary classification and a *predefined* sequential domain scheduler for the data partition during meta-train and -test stages.

However, these existing meta-learning-based OSDG approaches, *i.e.*, MEIDC [54] and MLDG [46], do not consider how the order in which domains are presented during training affects model gener-alization. We believe this overlooks the potential to dynamically adapt the domain scheduler used for data partition based on certain criteria, such as domain difficulty, which could result in a more targeted training strategy and, therefore, better outcomes. In this paper, we observe that different ordering strategies for domain presentation used for data partition during the meta-training and testing phases lead to significant variations in OSDG performance, emphasizing the critical role of domain scheduling in optimizing model generalization.

To bridge this gap, we introduce a new training strategy named the **E**vidential **Bi-L**evel **Ha**rdest **D**omain **S**cheduler (EBiL-HaDS), which allows *dynamically adjusting the order of domain presentation* during data partitioning in the meta-training and -testing phases. The key idea of our method is to quantify domain reliability, defined as the aggregated confidence of the model on the samples across unseen domains, which will then be used as the main criterion for the data partition. To assess the domain reliability, we incorporate a secondary follower network to assess the domain reliability alongside the primary network. This allows for prioritizing the optimization of meta-learning on less reliable domains, facilitating an adaptive domain scheduler-based data partitioning. This follower network is trained using bi-level optimization, which involves a hierarchical setup where the solution to a lower-level optimization problem (evaluating domain reliability) serves as a constraint in an upper-level problem (meta-learning objective). Optimization of the follower network is guided by confidence scores generated through our proposed max rebiased evidential learning method, which adjusts the confidence by amplifying the differences between the decision boundaries of different classes. As a result, the follower network can better quantify the reliability of each domain based on how distinct and consistent the classification boundaries are, improving the ability to generalize to unseen domains. EBiL-HaDS enhances cross-domain generalizability and differentiation of seen and unseen classes by prioritizing training on less reliable domains through adaptive domain scheduling.

Our experiments demonstrate the effectiveness of domain scheduling via EBiL-HaDS on three established datasets: PACS [31], DigitsDG [63], and OfficeHome [51], which span a variety of image classification tasks. Results demonstrate that EBiL-HaDS significantly improves model generalizability in open-set scenarios, enhancing domain generalization and the model's ability to distinguish between known and unknown categories in new domains. This performance surpasses that of both random and standard sequential domain scheduling methods during training, underscoring EBiL-HaDS's potential to advance current OSDG capabilities in deep learning.

## 2 Related Work

We simultaneously address two challenges: domain generalization and open-set recognition. Domain generalization is a task that expects a model to generalize well to unseen domains while leveraging

multiple seen domains for training [48, 52]. Open-set recognition, on the other hand, aims to reject unseen categories at test-time, *e.g.*, by delivering low confidence scores in such cases [16].

Domain generalization methods usually alleviate the domain gap with techniques such as data augmentation [55, 40, 65, 18, 64, 34, 35], contrastive learning [57, 24, 28, 43], domain adversarial learning [15], domain-specific normalization [44], and GANs-based methods [10, 32]. For open-set recognition, common approaches include logits calibration [3, 42], evidential learning [59, 53, 62, 2], reconstruction-based approaches [58, 22], GANs-based methods [29], and reciprocal point-based approaches [8, 9].

A considerable cluster of research utilizes source domains that encompass diverse categories for training, as highlighted in [14, 47, 4, 8, 36, 61], where each category poses unique domain generalization challenges. The primary focus of these methodologies is to improve domain generalizability, and they tend to allocate less attention to the complexities associated with open-set scenarios. In this setting, the categories involved in each source domain may not be the same. ODG-Net proposed by Bose *et al.* [4] leverages GAN to synthesize data from the merged training domains to improve cross-domain generalizability. SWAD proposed by Chen *et al.* [8] uses models averaged across various training epochs. Katsumata *et al.* [27] propose to use metric learning to get discriminative embedding space which benefits the open-set domain generalization. Wang *et al.* [54] introduce a new MEDIC model together with a new formalization of the open-set domain generalization protocols, where the source domains share the same categories defined as seen. This benchmark definition balances the impact of the model's open-set recognition and domain generalization performance in evaluation, which is adopted in our work.

Newer OSDG approaches show great promise of meta-learning strategies for improving cross-domain generalization [54, 46]. Yet, these works mainly utilize a fixed, sequential scheduling of source domains during training [54, 46]. Existing works in curriculum learning indicate that using a specific training order at the instance level can benefit the model performance on various tasks [56, 26, 17, 37, 20, 41, 28, 45]. However, existing curriculum learning approaches usually operate at the instance level (scheduling individual dataset instances within standard training) and are not designed for OSDG tasks, while we focus on domain-based scheduling by quantifying the domain difficulty in meta-learning. The influence of domain scheduling in the OSDG task remains unexplored. This paper, for the first time, examines the effects of guiding the meta-learning process with an *adaptive domain scheduler*, named EBiL-HaDS, which achieves data partition based on a domain reliability measure estimated by a follower network, trained in a bi-level manner with the supervision from the confidence score optimized by a novel max rebiased discrepancy evidential learning.

## 3 Method

The most challenging domain is chosen to perform data partitioning for the meta-task reservation during meta-learning. In our proposed EBiL-HaDS, we first utilize max rebiased discrepancy evidential learning (Sec. 3.1) to achieve more reliable confidence acquisition, which is subsequently used as the supervision for the reliability prediction of the follower network. During the training stage, our method makes use of two networks with identical architectures: one serves as the main feature extraction network, and the other functions as the follower network, aiming to assess the domain reliability. The follower network is optimized in a bi-level manner alongside the main network (Sec. 3.2). Hardest domain selection is accomplished by aggregating votes for samples from each domain for the randomly selected reserved classes using the follower network (Sec. 3.3).

To optimize domain scheduling, we first define the term domain reliability, as the degree to which data from a domain consistently aids in improving the model's accuracy and generalizability across unseen domains. An important step is therefore to adaptively rate the domain reliability during training. To achieve this, we employ two parallel networks: the main network used for feature extraction, and the follower network, which assesses the reliability of different domains based on the refined confidence metrics. This follower network plays a central role in our adaptive domain scheduling strategy: it employs a voting process to identify and select the most challenging domains – those that exhibit the least reliability according to its assessments. After selecting the hardest domain, the data is divided into two sets. One set includes data from more challenging domains outside the reserved classes and data from more reliable domains within the reserved classes, which together form the

meta-training set. The complementary partitions of the meta-training set are used as the meta-testing set. Both networks are simultaneously optimized through a bi-level training approach, meaning that the outcome of a lower-level optimization problem (evaluating domain reliability) is a constraint in an upper-level problem (meta-learning objective). The entire pipeline during training when using the proposed EBiL-HaDS is depicted in Alg. 1.

## 3.1 Max Rebiased Discrepancy Evidential Learning

The domain scheduler we propose leverages a follower network to ascertain reliability measurements, based on reliability evaluations conducted before the start of each epoch for all source domains. As such, confidence calibration is crucial for the main network's functionality. Evidential learning, which has been extensively applied across various domains such as action recognition [62] and image classification [25], effectively calibrates these confidence predictions. However, a notable limitation of evidential learning is its propensity for overfitting, leading to suboptimal performance [11].

To address these challenges, we propose to regularize the evidential learning by novel rebiased discrepancy maximization, which is employed for the confidence calibration of the main network to encourage diverse decision boundaries. This method involves training dual decision-making heads designed to exhibit rebiased maximized discrepancies. The aim is to foster the development of both informative and dependable decision-making capabilities within the leveraged deep learning model. Let $\mathbf{x}$ denote a batch of data used in training, $M_\alpha$ denote the feature extraction backbone, $R_{\theta_1}$ and $R_{\theta_2}$ denote the two rebiased layers, and $\mathcal{K}$ denote the Gaussian kernel to reproduce the Hilbert space. We first calculate the max rebiased discrepancy regularization by Eq. 1,

$$\mathcal{R}_{RB}(\mathbf{x}; \Theta) = \sum_{i \in \{1,2\}} \mathbb{E}\left[\mathcal{K}(R_{\theta_i}(M_\alpha(\mathbf{x})), R_{\theta_i}(M_\alpha(\mathbf{x})))\right] - 2 * \mathbb{E}[\mathcal{K}(R_{\theta_1}(M_\alpha(\mathbf{x})), R_{\theta_2}(M_\alpha(\mathbf{x})))].$$
(1)

We aim to maximize the above loss function to achieve the maximum discrepancy between the embeddings extracted from the two rebiased layers. This maximization encourages the learned evidence from the two layers to diverge from each other, thereby capturing open-set domain generalization cues from two different perspectives. Deep evidential learning is then applied to the conventional classification head, providing an additional constraint to achieve more reliable confidence calibration, as described in Eq. 2.

$$\mathcal{L}_{RBE}(\mathbf{y}, \mathbf{x}; \Theta) = \sum_{i \in \{1,2\}} \left[ \sum_{c=1}^{\mathcal{C}} [\mathbf{y}_c \left(\log S_i - \log(R_{\theta_i}\left(M_\alpha(\mathbf{x})\right)_c + 1)\right)] \right] - \mathcal{R}_{RB}(\mathbf{x}; \Theta) \quad (2)$$

where $S_i = \sum_{c=1}^{C}(Dir(p|R_{\theta_i}\left(M_\alpha(\mathbf{x})\right)_c + 1)$ denotes the strength of a Dirichlet distribution, $\mathbf{y}_c$ is the one-hot annotation of sample $\mathbf{x}$ from class $c$, $p$ is the predicted probability. The two rebiased layers are engineered to capture distinct evidence by employing max discrepancy regularization. By averaging the logits produced by the two prediction heads on the top of the two rebiased layers for the conventional classification on the seen categories, we can harvest the final estimated confidence score. This score is subsequently utilized to supervise the follower network, as elaborated in the following subsections.

## 3.2 Follower Network for Reliability Learning

To establish an adaptive domain scheduler for the OSDG task, the most straightforward approach would involve training a network to directly predict the sequence in which domains are employed during the training phase for sample selection. However, this method does not facilitate gradient computation, thereby preventing the direct optimization of the scheduler network.

In this work, we propose an alternative method where a follower network is trained to assess the reliability of each sample, utilizing predicted confidence scores derived from max discrepancy evidential learning as supervision. Throughout the training process, we employ samples from various domains to collectively assess reliability. Additionally, we utilize a follower network, denoted as $M_\beta$, which mirrors the architecture of the main network, but with classification heads replaced by one regression head. $\Theta$ indicates all the parameters in the main network, including the parameters from

**Algorithm 1** Training with Evidential Bi-Level Hardest Domain Scheduler.

---

**Require:** Known domains $\mathcal{D}$; Known classes $\mathcal{C}$; backbone $M_\alpha$; two rebiased layers $R_{\theta_1}$ and $R_{\theta_2}$; two heads $H_{\phi_1}$ and $H_{\phi_2}$; follower scheduling network $M_\beta$; weighted cross entropy $WCE$; mean squared error $MSE$.

1: **while** not converged **do**
2:     Randomly select two known classes $c_i, c_j \leftarrow \mathcal{C}$;
3:     Get the hardest domain $d^*$ using $M_\beta$ by Eq. 5; Select two domains from $d_i, d_j \leftarrow \mathcal{D}/\{d^*\}$;
4:     Sample data $\Omega_a, \Omega_b = \{\mathbf{x}^{[c_k, d_k]}|k \in [i,j]\}, \{\mathbf{x}^{[c_k, d_k]}|c_k \in \{\mathcal{C}/\{c_i, c_j\}, d_k \in \{d^*\}\}$;
5:     Construct meta-train set by $\Omega_{m-train} = \Omega_a \cup \Omega_b$;
6:     **Meta-train:**
7:     **for** $\mathbf{x}$ in $\Omega_{m-train}$ **do**;
8:       Extract rebiased embeddings $\mathbf{f}_1 = R_{\theta_1}(M_\alpha(\mathbf{x}))$ and $\mathbf{f}_2 = R_{\theta_2}(M_\alpha(\mathbf{x}))$;
9:       Obtain the max rebiased discrepancy evidential learning loss $L_{RBE}(\mathbf{x})$ using $\mathbf{f}_1$ and $\mathbf{f}_2$;
10:      Follower learning $L_{REG}(\mathbf{x}) = MSE(M_\beta(\mathbf{x}), \frac{1}{2}\sum_{k \in \{1,2\}} Conf(H_{\Phi_k}(\mathbf{f}_k)))$;
11:      Obtain classification loss $L_{CLS}(\mathbf{x}) = \sum_{k \in \{1,2\}}(WCE(H_{\Phi_k}(M_\alpha(\mathbf{x}))), \mathbf{y}, \omega)), \omega \leftarrow M_\beta(\mathbf{x})$, where $\mathbf{y}$ indicates the classification annotation;
12:     **end for**
13:     $L_{m-train} \leftarrow \sum_{\mathbf{x} \in \Omega_{m-train}}(L_{CLS}(\mathbf{x}) + L_{REG}(\mathbf{x}) + L_{RBE}(\mathbf{x}))$. Backpropagation and parameter update for the whole network;
14:     **Meta-test:**
15:     Sample data $\Omega_a^*, \Omega_b^* = \{\mathbf{x}^{[c_k, d_k]}|c_k \in \{c_i, c_j\}, d_k \in \{d^*\}\}, \{\mathbf{x}^{[c_k, d_k]}|c_k \in \mathcal{C}/\{c_i, c_j\}, d_k \in \{d_i, d_j\}\}$. Construct meta-test set $\Omega_{m-test} = \Omega_a^* \cup \Omega_b^*$;
16:     Obtain loss for meta-test $L_{m-test} \leftarrow \sum_{\mathbf{x} \in \{\Omega_{m-test}\}}(L_{CLS}(\mathbf{x}) + L_{REG}(\mathbf{x}) + L_{RBE}(\mathbf{x}))$;
17:     Back propagation and parameter update using $L_{all} = L_{m-test} + L_{m-train}$.
18: **end while**

---

the backbone, rebiased layers, and heads. We aim to solve the optimization task, as shown in Eq. 3.

$$\Theta^* = \arg\min_\Theta \mathcal{L}_m(M_\Theta(\mathbf{x}), \omega^* \leftarrow M_{\beta^*}(\mathbf{x})) \quad \text{subject to} \quad \beta^* = \arg\min_\beta L_f(M_\Theta(\mathbf{x}), M_\beta(\mathbf{x})), \quad (3)$$

where $\omega^*$ indicates the instance-wise reliability which serves as the weight for each instance during the loss calculation. Substituting the best response function $\beta^*(\Theta) = \arg\min_\beta L_f(M_\Theta(\mathbf{x}), M_\beta(\mathbf{x}))$ provides a single-level problem, as shown in Eq. 4.

$$\Theta^* = \arg\min_\Theta L_m(\Theta, \beta^*(\Theta)), \quad (4)$$

where $L_m$ denotes classification loss ($L_{CLS}$) and $L_{RBE}$. $L_f$ denotes the regression loss ($L_{REG}$).

### 3.3 Hardest Domain Scheduler during Training

We illustrate the details of the training procedure by the proposed domain scheduler in Alg. 1. We adopt the meta-training framework outlined by MLDG [46], integrating our proposed domain scheduler to facilitate the data partition of meta-tasks. In this approach, optimization is achieved using both the meta-train and -test sets, characterized by distinct data distributions. For each domain present in the training dataset, we sample a batch that encompasses the reserved categories. Subsequently, we identify the most challenging domain by determining which domain exhibits the lowest reliability under the selected seen categories. This procedure is accomplished by the calculation of the expected reliability as in Eq. 5.

$$d^* = \arg\min_d(\{\omega_d|d \in \mathcal{D}\}), \quad \omega_d = \min_{c \in \mathcal{C}^*}\left[\exp\left[1 + \sum_{i=1}^{N_c^*} \frac{(M_\beta(\mathbf{x}_i^{(c,d)}))}{N_c^*}\right] * (0.1 + \sigma * \gamma_d)\right], \quad (5)$$

where $d^*$ denotes the estimated hardest domain. $N_c^*$ and $\mathcal{C}^*$ denote the number of samples from domain $d$ and the number of selected known categories at the start of one epoch. $\mathcal{D}$ denotes the known domains used during the training procedure. $\mathbf{x}_i^{(c,d)}$ indicates the $i$-th sample from class $c$ and domain $d$. $\gamma_d$ indicates the schedule frequency for domain $d$ in the past training period, which considers the balance of different domains.

Table 1: Results (%) of PACS on ResNet18 [21]. The open-set ratio is 6:1.

| Method | Photo (P) | | | Art (A) | | | Cartoon (C) | | | Sketch (S) | | | Avg | | |
|---|---|---|---|---|---|---|---|---|---|---|---|---|---|---|---|
| | Acc | H-score | OSCR | Acc | H-score | OSCR | Acc | H-score | OSCR | Acc | H-score | OSCR | Acc | H-score | OSCR |
| OpenMax [3] | 95.56 | 92.48 | - | 83.68 | 69.61 | - | 78.61 | 64.36 | - | 70.89 | 50.67 | - | 82.19 | 69.28 | - |
| ERM [50] | **96.04** | 93.40 | **95.11** | 84.18 | 70.54 | 71.89 | 77.63 | 62.80 | 62.57 | 70.44 | 55.81 | 51.75 | 82.07 | 70.64 | 70.33 |
| ARPL [8] | 94.83 | **95.06** | 94.63 | 83.93 | 67.88 | 68.82 | 78.56 | 62.98 | 65.30 | 74.34 | 61.20 | 59.80 | 82.91 | 71.78 | 72.14 |
| MMLD [39] | 94.83 | 88.80 | 92.94 | 84.43 | 64.83 | 69.43 | 77.11 | 64.21 | 65.36 | 75.14 | 67.70 | 64.69 | 82.88 | 71.38 | 73.11 |
| RSC [23] | 94.43 | 88.37 | 91.38 | 83.36 | 70.27 | 73.55 | 78.09 | 65.13 | 66.15 | 77.16 | 52.98 | 62.31 | 83.26 | 69.19 | 73.35 |
| DAML [46] | 91.44 | 80.87 | 82.83 | 83.11 | 72.05 | 71.75 | 79.11 | 66.26 | 66.46 | 82.97 | 72.63 | 73.71 | 84.16 | 72.95 | 73.69 |
| MixStyle [65] | 95.23 | 82.02 | 88.99 | 86.18 | 70.62 | 72.57 | 78.92 | 63.23 | 63.81 | 80.34 | 71.90 | 72.07 | 85.17 | 71.94 | 74.36 |
| SelfReg [28] | 95.72 | 89.34 | 92.26 | 86.24 | 72.45 | 73.77 | 80.77 | 65.75 | 66.38 | 78.30 | 67.06 | 65.69 | 85.26 | 73.65 | 74.53 |
| MLDG [30] | 94.99 | 91.48 | 93.70 | 84.12 | 69.52 | 72.15 | 78.45 | 61.59 | 64.32 | 79.99 | 69.67 | 68.60 | 84.39 | 73.06 | 74.69 |
| MVDG [60] | 94.43 | 74.07 | 88.07 | **87.62** | 71.98 | 75.05 | 81.18 | 63.95 | 66.34 | 82.41 | 73.55 | 73.83 | 86.41 | 70.89 | 75.82 |
| ODG-Net [4] | 93.54 | 89.39 | 89.76 | 85.74 | 72.36 | 73.41 | 81.59 | 67.04 | 67.99 | 79.89 | 61.57 | 67.46 | 85.19 | 72.59 | 74.66 |
| MEDIC-cls [54] | 94.83 | 83.68 | 90.30 | 86.20 | 69.35 | 74.16 | 81.94 | 63.26 | 67.43 | 81.84 | 69.60 | 70.85 | 86.20 | 71.47 | 75.69 |
| MEDIC-bcls [54] | 94.83 | 89.49 | 92.40 | 86.20 | 73.82 | 75.58 | 81.94 | 66.26 | 69.04 | 81.84 | 74.37 | 74.52 | 86.20 | 75.98 | 77.89 |
| EBiL-HaDS-cls (ours) | 95.80 | 91.54 | 94.62 | 87.24 | 71.87 | 74.15 | **82.98** | **68.55** | 71.62 | 83.21 | 74.89 | 74.50 | **87.31** | 76.71 | 78.72 |
| EBiL-HaDS-bcls (ours) | 95.80 | 93.10 | 94.42 | 87.24 | **75.66** | **77.19** | **82.98** | 67.57 | **72.22** | 83.21 | **78.29** | **77.52** | **87.31** | **78.66** | **80.34** |

Table 2: Results (%) of PACS on ResNet50 [21]. The open-set ratio is 6:1.

| Method | Photo (P) | | | Art (A) | | | Cartoon (C) | | | Sketch (S) | | | Avg | | |
|---|---|---|---|---|---|---|---|---|---|---|---|---|---|---|---|
| | Acc | H-score | OSCR | Acc | H-score | OSCR | Acc | H-score | OSCR | Acc | H-score | OSCR | Acc | H-score | OSCR |
| OpenMax [3] | 97.58 | 93.09 | - | 88.37 | 73.91 | - | 84.38 | 68.23 | - | 80.07 | 68.06 | - | 87.60 | 75.82 | - |
| ARPL [8] | 97.09 | **96.81** | 96.86 | 88.24 | 77.48 | 80.32 | 82.68 | 67.19 | 68.31 | 78.08 | 70.04 | 69.47 | 86.52 | 77.88 | 78.74 |
| MIRO [7] | 94.85 | 92.32 | 93.27 | 88.51 | 65.02 | 79.01 | 82.98 | 63.05 | 73.72 | 82.22 | 69.47 | 70.61 | 87.14 | 72.47 | 79.15 |
| MLDG [30] | 96.77 | 95.85 | 96.33 | 87.99 | 77.16 | 79.93 | 83.45 | 68.74 | 71.32 | 82.25 | 73.16 | 72.27 | 87.61 | 78.73 | 79.96 |
| ERM [50] | 97.09 | 96.58 | 96.68 | 89.99 | 76.05 | 82.44 | 85.10 | 65.79 | 70.59 | 80.31 | 70.29 | 70.16 | 88.12 | 77.18 | 79.97 |
| CIRL [38] | 96.53 | 87.75 | 95.40 | 92.06 | 70.75 | 77.44 | 85.71 | 68.82 | 73.71 | 84.35 | 66.73 | 77.24 | 89.66 | 73.51 | 80.95 |
| MixStyle [65] | 96.53 | 93.57 | 95.30 | 90.87 | 79.15 | 83.27 | 86.80 | 68.08 | 74.68 | 84.88 | 71.57 | 73.41 | 89.77 | 78.09 | 81.66 |
| CrossMatch [67] | 96.53 | 96.34 | 96.12 | 91.37 | 75.67 | 82.32 | 83.92 | 67.02 | 74.55 | 81.61 | 72.03 | 73.99 | 88.37 | 77.76 | 81.75 |
| SWAD [6] | 96.37 | 84.56 | 93.24 | 93.75 | 68.41 | 85.00 | 85.57 | 58.57 | 75.90 | 81.90 | 74.66 | 74.65 | 89.40 | 71.55 | 82.20 |
| MVDG [60] | 97.17 | 95.02 | 96.63 | **92.50** | 79.47 | 85.02 | 86.02 | 71.05 | 76.03 | 83.44 | 75.24 | 75.18 | 89.78 | 80.20 | 83.21 |
| ODG-Net [4] | 96.53 | 94.93 | 95.58 | 89.24 | 65.22 | 74.60 | 83.86 | 64.32 | 71.20 | 84.80 | 77.58 | 77.38 | 88.61 | 75.51 | 79.69 |
| MEDIC-cls [54] | 96.37 | 93.80 | 95.37 | 91.62 | 80.80 | 84.75 | 86.65 | 75.85 | 77.48 | 84.61 | 75.80 | 76.79 | 89.81 | 81.56 | 83.58 |
| MEDIC-bcls [54] | 96.37 | 94.75 | 95.79 | 91.62 | 81.61 | 85.81 | 86.65 | 77.39 | 78.30 | 84.61 | 78.35 | 79.50 | 89.81 | 83.03 | 84.85 |
| EBiL-HaDS-cls (ours) | **97.82** | 93.58 | 95.69 | 92.31 | 80.95 | 84.35 | **87.52** | 75.68 | 78.68 | **85.91** | 76.05 | 78.57 | **90.89** | 81.57 | 84.32 |
| EBiL-HaDS-bcls (ours) | **97.82** | 96.04 | **97.14** | 92.31 | **82.80** | **86.17** | **87.52** | **78.34** | **79.85** | **85.91** | **78.68** | **81.32** | **90.89** | **83.97** | **86.12** |

## 4 Experiments

### 4.1 Implementation Details

All the experiments use PyToch 2.0 and one NVIDIA A100 GPU. We set the upper limit of the training step as $1e^4$ and use SGD optimizer, where the learning rate (lr) is set as $1e^{-3}$ and batch size is chosen as 16. The weights of $L_{CLS}$, $L_{REG}$, and $L_{RBE}$ are chosen as 1.0, $1e^{-4}$, and $5e^{-4}$. Lr decay is $1e^{-1}$ and conducted at $8e^3$ meta-training step. The worker number is 4 and $\gamma$ is $2e^{-5}$. For ResNet18 [21] and ResNet50 [21], each rebiased layer is constructed using a residual convolutional block. For ConvNet [66], convolutional layers are utilized. Apart from the conventional classification head (cls), we also utilize a binary classification head (bcls) as in MEDIC [54]. The training time of our method is 1h on PACS (ResNet18 [21]), 1.2h on PACS (ResNet50 [21]), 20min on DigitsDG (ConvNet [66]), 2h on OfficeHome (ResNet18 [21]). The parameter ablation is provided in the appendix.

### 4.2 Datasets and Metrics

We adopt the open-set protocols provided by MEDIC [54], wherein the training set of each domain, shares the same categories. Three benchmarks are involved. **PACS** [31] comprises 4 distinct domains, *i.e.*, *photo*, *art-painting*, *cartoon*, and *sketch*, totaling 9,991 images. 7 classes are contained in this dataset for each domain. **Digits-DG** [63] aggregates 4 standard digit recognition dataset, *i.e.*, *Mnist*, *Mnist-m*, *SVHN*, and *SYN*. **Office-Home** [51] includes 15,500 images across 65 classes from 4 domains, *i.e.*, *art*, *clipart*, *product*, and *real-world*. Since MEDIC [54] did not provide a detailed benchmark on OfficeHome, we construct the whole benchmark using several outstanding baselines and our approach, where the last 30 categories following the alphabet order are chosen as unseen categories. The leave-one-domain-out DG setting is adopted. Close-set accuracy (acc), H-score, and OSCR serve as metrics following [54], where OSCR is the primary metric for OSDG.

Table 3: Results (%) of PACS on ResNet152 [21]. The open-set ratio is 6:1.

| Method | Photo (P) | | | Art (A) | | | Cartoon (C) | | | Sketch (S) | | | Avg | | |
|---|---|---|---|---|---|---|---|---|---|---|---|---|---|---|---|
| | Acc | H-score | OSCR | Acc | H-score | OSCR | Acc | H-score | OSCR | Acc | H-score | OSCR | Acc | H-score | OSCR |
| ARPL | 94.35 | 85.45 | 86.74 | 89.81 | 71.27 | 78.53 | 83.91 | 69.75 | 72.08 | 77.53 | 52.70 | 66.68 | 77.53 | 69.81 | 76.01 |
| MLDG | 96.20 | 91.07 | 94.64 | 89.81 | 77.65 | 82.19 | 83.86 | 73.66 | 74.03 | 82.89 | 64.30 | 72.98 | 88.19 | 76.67 | 80.96 |
| SWAD | 95.64 | 84.82 | 89.74 | 86.30 | 73.86 | 75.91 | 78.49 | 70.18 | 68.41 | 76.92 | 75.33 | 63.35 | 84.34 | 76.05 | 74.35 |
| ODG-Net | 95.88 | 89.11 | 91.85 | 89.62 | 80.65 | 82.48 | 85.15 | 70.37 | 73.66 | 79.30 | 77.00 | 72.22 | 87.49 | 79.28 | 80.05 |
| MEDIC-cls | 94.67 | 49.54 | 76.98 | 89.37 | 73.26 | 77.79 | 86.59 | 68.49 | 74.82 | 85.81 | 56.14 | 78.83 | 89.11 | 61.86 | 77.11 |
| MEDIC-bcls | 94.67 | 72.88 | 81.30 | 89.37 | 74.92 | 78.70 | 86.59 | 71.46 | 75.17 | 85.81 | 58.80 | 78.32 | 89.11 | 69.52 | 78.37 |
| EBiL-HaDS-cls (ours) | 97.90 | 91.66 | 96.62 | 92.06 | 81.52 | 85.43 | 87.21 | 76.61 | 78.19 | 87.08 | 81.13 | 80.21 | 91.06 | 82.73 | 85.11 |
| EBiL-HaDS-bcls (ours) | 97.90 | 94.34 | 97.39 | 92.06 | 82.00 | 85.94 | 87.21 | 76.62 | 80.15 | 87.08 | 88.57 | 81.52 | 91.06 | 85.38 | 86.25 |

Table 4: Results (%) of PACS on ViT base model [13] (patch size 16 and image size 224). The open-set ratio is 6:1.

| Method | Photo (P) | | | Art (A) | | | Cartoon (C) | | | Sketch (S) | | | Avg | | |
|---|---|---|---|---|---|---|---|---|---|---|---|---|---|---|---|
| | Acc | H-score | OSCR | Acc | H-score | OSCR | Acc | H-score | OSCR | Acc | H-score | OSCR | Acc | H-score | OSCR |
| ARPL | 99.19 | 95.31 | 98.61 | 90.49 | 85.46 | 88.59 | 81.88 | 72.17 | 73.34 | 63.01 | 29.33 | 50.59 | 83.64 | 70.57 | 77.78 |
| MLDG | 99.19 | 95.40 | 98.88 | 91.87 | 82.46 | 89.47 | 80.56 | 69.62 | 74.19 | 61.66 | 40.79 | 43.88 | 83.32 | 72.07 | 76.61 |
| SWAD | 98.55 | 93.19 | 97.62 | 90.81 | 81.34 | 88.52 | 83.24 | 73.03 | 76.59 | 57.89 | 35.83 | 41.68 | 82.62 | 70.85 | 76.10 |
| ODG-Net | 97.58 | 96.24 | 95.23 | 90.49 | 83.32 | 87.90 | 82.36 | 68.66 | 75.80 | 62.59 | 43.59 | 50.22 | 83.26 | 72.95 | 77.29 |
| MEDIC-cls | 99.03 | 95.33 | 98.22 | 92.06 | 83.27 | 87.46 | 85.62 | 69.79 | 75.37 | 68.40 | 41.95 | 56.56 | 86.28 | 72.59 | 79.40 |
| MEDIC-bcls | 99.03 | 96.04 | 97.55 | 92.06 | 82.68 | 87.73 | 85.62 | 69.15 | 76.80 | 68.40 | 39.60 | 55.92 | 86.28 | 71.87 | 79.50 |
| EBiL-HaDS-cls (ours) | 99.52 | 97.30 | 99.11 | 94.68 | 86.10 | 92.10 | 89.22 | 74.31 | 77.76 | 69.49 | 44.34 | 55.37 | 88.23 | 75.53 | 81.09 |
| EBiL-HaDS-bcls (ours) | 99.52 | 96.91 | 99.18 | 94.68 | 88.31 | 92.28 | 89.22 | 73.91 | 77.95 | 69.49 | 48.09 | 56.78 | 88.23 | 76.81 | 81.55 |

Table 5: Results (%) of Digits-DG on ConvNet [66]. The open-set ratio is 6:4.

| Method | MNIST | | | MNIST-M | | | SVHN | | | SYN | | | Avg | | |
|---|---|---|---|---|---|---|---|---|---|---|---|---|---|---|---|
| | Acc | H-score | OSCR | Acc | H-score | OSCR | Acc | H-score | OSCR | Acc | H-score | OSCR | Acc | H-score | OSCR |
| OpenMax [3] | 97.33 | 52.03 | - | 71.03 | 57.26 | - | 72.00 | 49.46 | - | 84.83 | 54.78 | - | 81.30 | 53.38 | - |
| MixStyle [65] | 97.86 | 73.25 | 89.36 | 74.50 | 59.30 | 56.95 | 69.28 | 53.24 | 48.43 | 85.06 | 62.06 | 65.44 | 81.68 | 61.50 | 65.05 |
| ERM [50] | 97.47 | 80.90 | 92.60 | 71.03 | 53.92 | 54.04 | 71.08 | 54.37 | 49.86 | 85.67 | 51.57 | 67.63 | 81.31 | 60.19 | 66.03 |
| ARPL [8] | 97.75 | 85.74 | 91.86 | 69.78 | 58.08 | 54.21 | 71.78 | 56.98 | 53.63 | 85.31 | 64.04 | 65.89 | 81.16 | 66.21 | 66.40 |
| MLDG [30] | 97.83 | 80.36 | 94.28 | 71.11 | 46.84 | 55.17 | 73.64 | 53.54 | 53.64 | 86.08 | 63.56 | 70.34 | 82.16 | 61.08 | 68.36 |
| SWAD [6] | 97.71 | 84.44 | 92.65 | 73.09 | 53.35 | 55.94 | 76.08 | 59.18 | 56.25 | 87.95 | 51.27 | 69.03 | 83.71 | 62.06 | 68.47 |
| ODG-Net [4] | 96.86 | 71.34 | 90.93 | 72.92 | 58.47 | 56.98 | 69.83 | 55.74 | 51.55 | 85.42 | 67.67 | 68.12 | 81.26 | 63.31 | 66.90 |
| MEDIC-cls [54] | 97.89 | 67.37 | 96.17 | 71.14 | 48.44 | 55.37 | 76.00 | 51.20 | 55.58 | 88.11 | 64.94 | 73.62 | 83.28 | 57.98 | 70.19 |
| MEDIC-bcls [54] | 97.89 | 83.20 | 95.81 | 71.14 | 60.98 | 58.28 | 76.00 | 58.77 | 57.60 | 88.11 | 62.24 | 72.91 | 83.28 | 66.30 | 71.15 |
| EBiL-HaDS-cls | 99.50 | 87.40 | 97.49 | 74.28 | 56.58 | 60.86 | 80.33 | 61.27 | 62.84 | 93.97 | 73.95 | 78.14 | 87.02 | 70.27 | 75.83 |
| EBiL-HaDS-bcls | 99.50 | 91.63 | 97.58 | 74.28 | 60.72 | 59.39 | 80.33 | 62.23 | 63.88 | 93.97 | 69.92 | 79.28 | 87.02 | 71.13 | 75.87 |

Table 6: Results (%) of OfficeHome on ResNet18 [21]. The open-set ratio is 35:30.

| Method | Art | | | Clipart | | | Real World | | | Product | | | Avg | | |
|---|---|---|---|---|---|---|---|---|---|---|---|---|---|---|---|
| | Acc | H-score | OSCR | Acc | H-score | OSCR | Acc | H-score | OSCR | Acc | H-score | OSCR | Acc | H-score | OSCR |
| OpenMax [3] | 65.59 | 56.00 | - | 60.02 | 47.34 | - | 83.56 | 70.48 | - | 80.50 | 68.45 | - | 72.92 | 60.57 | - |
| MixStyle [65] | 62.81 | 53.93 | 50.71 | 52.46 | 44.53 | 42.27 | 81.16 | 67.70 | 67.95 | 76.29 | 63.46 | 62.51 | 68.18 | 57.41 | 55.86 |
| ERM [50] | 66.30 | 57.39 | 54.86 | 59.60 | 46.81 | 47.84 | 84.50 | 69.99 | 74.03 | 80.81 | 67.44 | 67.44 | 72.80 | 60.40 | 61.04 |
| ARPL [8] | 60.06 | 50.34 | 45.68 | 54.82 | 45.72 | 43.21 | 76.24 | 62.04 | 61.73 | 75.30 | 62.47 | 60.19 | 66.61 | 55.14 | 52.70 |
| MLDG [30] | 66.56 | 52.45 | 55.10 | 58.85 | 53.09 | 47.69 | 80.10 | 70.66 | 70.02 | 75.02 | 66.16 | 63.49 | 70.13 | 60.59 | 59.08 |
| SWAD [6] | 59.12 | 53.05 | 47.87 | 57.37 | 45.78 | 47.28 | 78.38 | 66.43 | 65.48 | 76.50 | 64.29 | 63.28 | 67.84 | 57.39 | 58.95 |
| ODG-Net [4] | 64.10 | 54.97 | 50.64 | 61.06 | 52.26 | 48.33 | 83.93 | 70.04 | 71.34 | 79.07 | 65.47 | 65.49 | 72.04 | 60.69 | 58.95 |
| MEDIC-cls [54] | 66.81 | 55.78 | 55.85 | 61.14 | 54.21 | 48.51 | 85.03 | 71.16 | 73.15 | 80.69 | 67.72 | 68.09 | 73.42 | 62.22 | 61.40 |
| MEDIC-bcls [54] | 66.81 | 51.76 | 56.21 | 61.14 | 53.28 | 48.97 | 85.03 | 70.61 | 74.08 | 80.69 | 67.17 | 67.17 | 73.42 | 60.82 | 61.61 |
| EBiL-HaDS-cls | 68.18 | 59.66 | 56.83 | 63.48 | 57.01 | 52.26 | 85.48 | 72.88 | 74.45 | 81.61 | 71.03 | 70.25 | 74.69 | 65.15 | 63.45 |
| EBiL-HaDS-bcls | 68.18 | 53.57 | 57.49 | 63.48 | 52.12 | 53.14 | 85.48 | 74.20 | 75.64 | 81.61 | 72.20 | 71.62 | 74.69 | 62.97 | 64.47 |

## 4.3 Analysis of the Experimental Results on Three OSDG Benchmarks

We first validate the performances of our approach on the three well-established benchmarks for the OSDG task, *e.g.*, PACS, DigitsDG, and OfficeHome. In Table 1, we use ResNet18 [21] as the feature extraction backbone which is pre-trained on ImageNet21K [12] for the PACS benchmark. Compared with the state-of-the-art methods, *i.e.*, MEDIC and ODG-Net, the proposed EBiL-HaDS achieves promising performance improvements for all the domain generalization splits. On averaged metrics across all of these DG splits, our approach delivers performance improvements by $1.11\%$ of close-set accuracy, $2.68\%$ of H-score, and $2.45\%$ of OSCR for the binary classification head (bcls), and consistent performance improvements can be found in the conventional classification head (cls). Through using EBiL-HaDS to achieve a more reasonable domain scheduler during the training, we observe that on the most challenging domain generalization split, *i.e.*, *Cartoon* as unseen domain, EBiL-HaDS delivers the most performance benefits. The core strength of EBiL-HaDS is its adaptive domain scheduling, optimized through a bi-level manner to achieve maximum rebiased discrepancy evidential learning. This ensures comprehensive and discriminative data partitions during

Table 7: Module ablation of the DigitsDG on ConvNet [66].

| Head | DGS | RBE | MNIST | | | MNIST-M | | | SVHN | | | SYN | | | Avg | | |
|---|---|---|---|---|---|---|---|---|---|---|---|---|---|---|---|---|---|
| | | | Acc | H-score | OSCR | Acc | H-score | OSCR | Acc | H-score | OSCR | Acc | H-score | OSCR | Acc | H-score | OSCR |
| cls | | | 97.89 | 67.37 | 96.17 | 71.14 | 48.44 | 55.37 | 76.00 | 51.20 | 55.58 | 88.11 | 64.90 | 73.62 | 83.28 | 57.98 | 70.19 |
| bcls | | | 97.89 | 83.20 | 95.81 | 71.14 | **60.98** | 58.28 | 76.00 | 58.77 | 57.60 | 88.11 | 62.24 | 72.91 | 83.28 | 66.30 | 71.15 |
| cls | ✓ | | 99.17 | 79.09 | 95.37 | 71.78 | 60.66 | 57.05 | 78.58 | 60.34 | 59.52 | 91.28 | 70.05 | 76.65 | 85.20 | 67.54 | 72.15 |
| bcls | ✓ | | 99.17 | 80.52 | 96.08 | 71.78 | 55.79 | 58.10 | 78.52 | 61.51 | 60.89 | 91.28 | 72.31 | 74.55 | 85.20 | 67.35 | 72.41 |
| cls | | ✓ | 99.14 | 79.47 | 95.14 | 72.06 | 59.19 | 56.76 | 77.61 | 56.88 | 58.37 | 91.11 | 72.28 | 72.78 | 84.98 | 66.96 | 70.76 |
| bcls | | ✓ | 99.14 | 81.03 | 96.03 | 72.06 | 60.89 | 57.60 | 77.61 | 59.21 | 59.32 | 91.11 | 73.53 | 73.96 | 84.98 | 68.67 | 71.98 |
| cls | ✓ | ✓ | **99.50** | 87.40 | 97.49 | **74.28** | 56.58 | **60.86** | **80.33** | 61.27 | 62.84 | **93.97** | 75.82 | 78.14 | **87.02** | 70.27 | 75.83 |
| bcls | ✓ | ✓ | **99.50** | 91.63 | 97.58 | **74.28** | 60.72 | 59.39 | **80.33** | 62.23 | 63.88 | **93.97** | 75.77 | **79.28** | **87.02** | 72.59 | 75.87 |

training, enhancing generalization to unseen domains, which is observed from the above experimental analyses. Significant performance gains in challenging DG splits, such as the unseen Cartoon domain, demonstrate its effectiveness in handling extreme domain shifts. Consistent metric improvements highlight EBiL-HaDS's versatility across various OSDG challenges.

Further experiments on a different backbone, *i.e.*, ResNet50 [21], are delivered in Table 2, where our method contributes 1.08%, 0.94%, and 1.27% performance improvements of close-set accuracy, H-score, and OSCR for binary classification head and consistent performance improvements for the conventional classification head. EBiL-HaDS contributes more performance improvements when we compare the experimental results on ResNet18 with ResNet50 [21] for the PACS dataset, which illustrates that the EBiL-HaDS is more helpful in alleviating the generalizability issue of model-preserving light-weight network structure since network with small size is hard to optimize and obtain the generalizable capabilities on challenging unseen domains.

We further conduct ablation on model architecture on ResNet152 and ViT base model [13], where our proposed method is compared with the MEDIC and other challenging baselines, *i.e.*, ARPL, MLDG, SWAD, and ODG-Net. From Table 2 and Table 1 we can observe an obvious OSDG performance improvement by increasing the complexity of the leveraged feature extraction backbone. However, when we compare Table 3 and Table 2, some baseline approaches trained with ResNet152 backbone even show performance decay on the major evaluation metric OSCR. This observation demonstrates that most OSDG methods face the overfitting issue when using a very large backbone, which is a critical issue for open-set challenging to recognize samples from unseen categories, especially in an unseen domain. The MEDIC-bcls approach shows 6.48% performance degradation on OSCR when we replace the backbone from ResNet50 to ResNet152. Using the proposed BHiL-HaDS to achieve a more reasonable task reservation in meta-learning procedure, BHiL-HaDS-bcls delivers 86.25% in terms of OSCR on ResNet152 [21] backbone, where the OSCR performance of BHiL-HaDS-bcls on ResNet50 is 86.12%.

This observation shows that our proposed adaptive domain scheduler can make the meta-learning effective on large complex models by reserving reasonable task for model optimization. Additional experiments on the ViT base model [13] (patch size 16 and window size 224) are provided in Table 4, where we observe that our proposed method can deliver consistent performance gains on the transformer architecture.

This observation is further validated by the experimental results on DigitsDG where a smaller network structure, *i.e.*, ConvNet [66], is used, as shown in Table 5. Our method contributes 3.74%, 4.83%, and 4.72% performance improvements of close-set accuracy, H-score, and OSCR for binary classification head and 3.74%, 12.29%, and 5.64% performance improvements of close-set accuracy, H-score, and OSCR for the conventional classification head. Consistent performance improvements are shown in Table 6 on the OfficeHome. We further observe that using EBiL-HaDS, the optimized model can contribute a distinct separation between confidence scores of the model on the unseen categories and seen categories in the test unseen domain, showing the benefits of a reasonable domain scheduler for OSDG.

## 4.4 Analysis of the Ablation Experiments

We deliver the ablation experiments in Table 7, We first remove the $L_{RBE}$ by directly supervising the follower network using the confidence score provided by SoftMax supervised by cross-entropy loss for classification, where the results are shown in the second part of Table 7 (w/o RBE). Compared with this ablation, our method achieves 1.82%, 5.24%, and 3.46% performance improvements of

Table 8: Comparison with different domain schedulers on DigitsDG with open-set ratio 6:4.

| Method | MNIST Acc | MNIST H-score | MNIST OSCR | MNIST-M Acc | MNIST-M H-score | MNIST-M OSCR | SVHN Acc | SVHN H-score | SVHN OSCR | SYN Acc | SYN H-score | SYN OSCR | Avg Acc | Avg H-score | Avg OSCR |
|---|---|---|---|---|---|---|---|---|---|---|---|---|---|---|---|
| SequentialSched-cls | 97.89 | 67.37 | 96.17 | 71.14 | 48.44 | 55.37 | 76.00 | 51.20 | 55.58 | 88.11 | 64.90 | 73.62 | 83.28 | 57.98 | 70.19 |
| SequentialSched-bcls | 97.89 | 83.20 | 95.81 | 71.14 | **60.98** | 58.28 | 76.00 | 58.77 | 57.60 | 88.11 | 62.24 | 72.91 | 83.28 | 66.30 | 71.15 |
| Random-cls | 98.39 | 52.93 | 94.21 | 70.92 | 52.70 | 52.41 | 77.92 | 59.65 | 57.95 | 88.33 | 44.11 | 75.66 | 83.89 | 52.35 | 70.06 |
| Random-bcls | 98.39 | 73.67 | 94.22 | 70.92 | 57.23 | 54.87 | 77.92 | 57.54 | 61.06 | 88.33 | 68.34 | 74.81 | 83.89 | 64.20 | 71.24 |
| EBiL-HaDS-cls | **99.50** | 87.40 | 97.49 | **74.28** | 56.58 | **60.86** | 80.33 | 61.27 | 62.84 | **93.97** | 75.82 | 78.14 | **87.02** | 70.27 | 75.83 |
| EBiL-HaDS-bcls | **99.50** | **91.63** | **97.58** | **74.28** | 60.72 | 59.39 | 80.33 | **62.23** | **63.88** | **93.97** | 75.77 | **79.28** | **87.02** | **72.59** | **75.87** |

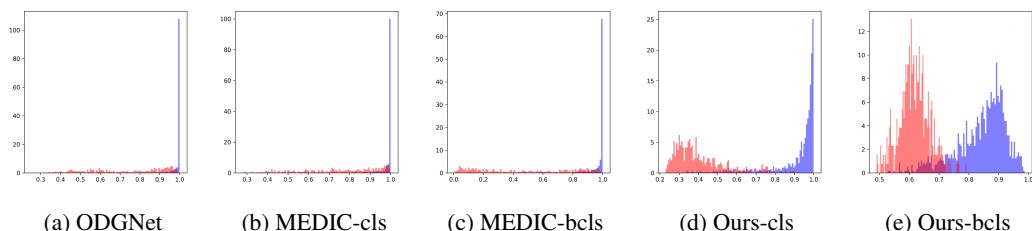

(a) ODGNet (b) MEDIC-cls (c) MEDIC-bcls (d) Ours-cls (e) Ours-bcls

Figure 1: Comparison of open-set confidence using ResNet18 [21] on PACS. *Photo* is the unseen domain. We use red and blue colors to denote unseen and seen categories.

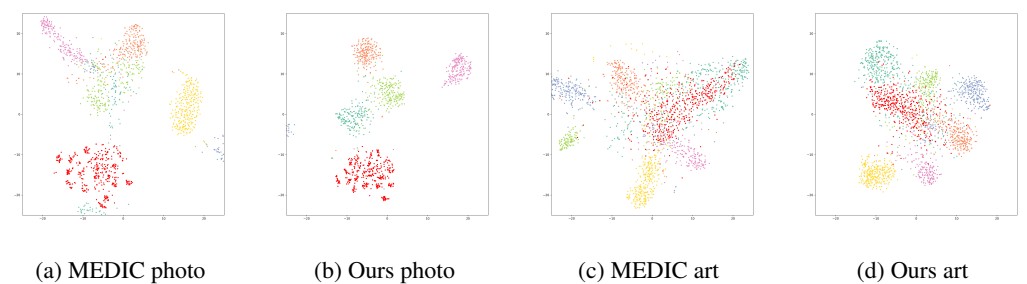

(a) MEDIC photo (b) Ours photo (c) MEDIC art (d) Ours art

Figure 2: The visualization of the embeddings through TSNE [49] for PACS on ResNet18 [21].

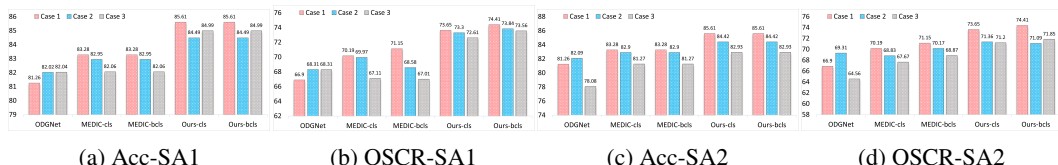

(a) Acc-SA1 (b) OSCR-SA1 (c) Acc-SA2 (d) OSCR-SA2

Figure 3: Ablation of different open-set ratios on DigitsDG dataset by ConvNet [66] backbone, where SA indicates Split Ablation. Regarding all the splits, Case 1 (denoted by red color) indicates using 7, 8, 9, 10 as unseen categories. In (a) and (b), Case 2 (denoted by blue color) and Case 3 (denoted by gray color) indicate that using 0, 1, 2, 3 and 2, 3, 4, 5 as unseen categories. In (c) and (d), Case 2 and Case 3 indicate using 7, 8, 9 and 8, 9 as unseen categories, respectively.

close-set accuracy, H-score, and OSCR for binary classification head and $1.82\%$, $2.73\%$, and $3.68\%$ performance improvements of close accuracy, H-score, and OSCR for conventional classification head. The significant OSDG performance improvements highlight the importance of the confidence score learned by the max rebiased discrepancy evidential learning in supervising the follower network, ensuring the promising reliability prediction. Then we use a sequential scheduler and keep the $L_{RBE}$ in the third part of Table 7 (w/o DGS), where our approach outperforms this variant by $2.04\%$, $3.92\%$, and $3.89\%$ of close accuracy, H-score, and OSCR for binary classification head and $2.04\%$, $3.31\%$, and $5.07\%$ of these metrics for conventional classification head. This observation shows the importance of the proposed domain scheduler for the OSDG task and highlights the effect of using meta-learning trained with a reasonable data partition. Both ablations show better OSDG performances compared with MEDIC, confirming the benefit of each component. We further deliver

more ablations, *e.g.*, the benefits brought by the max discrepancy regularization term, comparison with recent curriculum learning approaches, and ablation of the rebiased layers in the appendix.

### 4.5 Comparison of Different Domain Schedulers for OSDG Task

We present several comparison experiments in Table 8 to demonstrate the efficacy of various domain schedulers when applying meta-learning to the OSDG task. We compare our proposed EBiL-HaDS with both the sequential domain scheduler and the random domain scheduler. The sequential domain scheduler selects domains in a fixed order for batch data partitioning, while the random scheduler assigns domains randomly. Results show that EBiL-HaDS significantly outperforms both the random and sequential domain schedulers. Specifically, EBiL-HaDS achieves performance improvements of 3.13%, 8.39%, and 4.63% in closed accuracy, H-score, and OSCR for the binary classification head, and 3.13%, 17.92%, and 5.77% in these metrics for the conventional classification head compared to the random scheduler. This ablation demonstrates that our scheduler enables the model to converge to a more optimal region, which outperforms both predefined fixed-order (sequential) and maximally random (random) schedulers, underscoring the importance of a well-designed domain scheduler in meta-learning for the OSDG. More ablations on domain schedulers are supplemented in the appendix.

### 4.6 Analysis of the TSNE Visualizations of the Latent Space

In Figure 2, we deliver the TSNE [49] visualization of the latent space of MEDIC and our approach on the OSDG splits, *i.e.*, *photo* and *art* as unseen domains. Unseen and seen categories are denoted by red and other colors. we observe that the model trained by our method delivers a more compact cluster for each category and the unseen category is more separable regarding the decision boundary in the latent space. Our method's ability to improve the generalizability of the model is particularly noteworthy. The well-structured latent space facilitates better transfer learning capabilities, allowing the model to adapt more efficiently to new, unseen categories. This characteristic is especially beneficial in dynamic environments where the data distribution can change over time. In essence, the effectiveness of our approach in achieving a more discriminative and generalizable latent space can be directly linked to the sophisticated data partitioning achieved through EBiL-HaDS. This demonstrates the profound influence that carefully designed domain schedulers can have on the overall performance of deep learning models, emphasizing the need for thoughtful consideration in their implementation.

### 4.7 Ablation of the Open-Set Ratios and the Number of Unseen Categories

We first conduct the ablation towards different unseen categories with a predefined open-set ratio in Figure 3a and Figure 3b of close-set accuracy and the OSCR for open-set evaluation, where the performance of the ODG-NET, binary classification head, and conventional classification head of MEDIC method and our method are presented. We then conduct the ablation towards different numbers of unseen categories in Figure 3c and Figure 3d of close-set accuracy and the OSCR for open-set evaluation, where the performance of the ODG-NET, binary classification head, and classification head of MEDIC method and our method are presented. From the experimental results and comparisons, we can find consistent performance improvements, indicating the high generalizability of our approach across different open-set ratios and unseen category settings.

## 5 Conclusion

In this study, we introduce the Evidential Bi-Level Hardest Domain Scheduler (EBiL-HaDS) for the OSDG task. EBiL-HaDS is designed to create an adaptive domain scheduler that dynamically adjusts to varying domain difficulties. Extensive experiments on diverse image recognition tasks across three OSDG benchmarks demonstrate that our proposed solution generates more discriminative embeddings. Additionally, it significantly enhances the performance of state-of-the-art techniques in OSDG, showcasing its efficacy and potential for broader applications for deep learning models.

**Limitations and Societal Impacts.** EBiL-HaDS positively impacts society by enhancing model awareness of out-of-distribution categories in unseen domains, leading to more reliable decisions. It emphasizes the importance of domain scheduling in OSDG. However, the method may still result in misclassification and biased predictions, potentially causing negative effects. EBiL-HaDS relies on source domains with unified categories and has so far only been tested on image classification.

## Acknowledgements

The project served to prepare the SFB 1574 Circular Factory for the Perpetual Product (project ID: 471687386), approved by the German Research Foundation (DFG, German Research Foundation) with a start date of April 1, 2024. This work was also partially supported by the SmartAge project sponsored by the Carl Zeiss Stiftung (P2019-01-003; 2021-2026). This work was performed on the HoreKa supercomputer funded by the Ministry of Science, Research and the Arts Baden-Württemberg and by the Federal Ministry of Education and Research. The authors also acknowledge support by the state of Baden-Württemberg through bwHPC and the German Research Foundation (DFG) through grant INST 35/1597-1 FUGG. This project is also supported by the National Natural Science Foundation of China under Grant No. 62473139. Lastly, the authors thank for the support of Dr. Sepideh Pashami, the Swedish Innovation Agency VINNOVA, the Digital Futures.

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

# Appendix

## A  Further Illustration of the Evaluation Methods and Protocols

We follow the same protocol according to the MEDIC approach [54]. For PACS dataset which is used in Table 1 and Table 2, we use the open-set ratio as $6:1$, where the *elephant*, *horse*, *giraffe*, *dog*, *guitar*, *house* are selected as seen categories and *person* is selected as the unseen category. For the DigitsDG dataset, we leverage the open-set ratio as $6:4$, where digits $0, 1, 2, 3, 4, 5$ are used as seen categories while digits $6, 7, 8, 9$ are selected as unseen categories, in Table 5. For the OfficeHome dataset, *Mop*, *Mouse*, *Mug*, *Notebook*, *Oven*, *Pan*, *PaperClip*, *Pen*, *Pencil*, *PostitNotes*, *Printer*, *PushPin*, *Radio*, *Refrigerator*, *Ruler*, *Scissors*, *Screwdriver*, *Shelf*, *Sink*, *Sneakers*, *Soda*, *Speaker*, *Spoon*, *TV*, *Table*, *Telephone*, *ToothBrush*, *Toys*, *TrashCan*, *Webcam* are chosen as the 30 unseen categories. The *Acc* indicates the close-set accuracy measured on the seen categories to assess the correctness of the classification. The *H-score* and *OSCR* are measurements for the open-set recognition which are widely used in the OSDG field. Since the *H-score* relies on a predefined threshold derived from the source domain validation set to separate the seen categories and unseen categories, it is regarded as a secondary metric in our evaluation. MEDIC proposes OSCR for the OSDG evaluation where no predefined threshold is required, which is used as our primary evaluation metric.

Regarding the calculation of the H-Score, we first have a threshold ratio $\lambda$ to separate the samples coming from seen and unseen classes. When the predicted confidence score is below $\lambda$, we regard the corresponding samples as an unseen category. Then, we calculate the accuracy for all the samples regarded as seen categories according to their corresponding seen labels, which can be denoted as $Acc_k$. The accuracy calculation of the unseen categories is conducted in a binary classification manner, where the label for the samples from the seen category is annotated as $1$ and the label for the samples from the unseen category is annotated as $0$. Then the accuracy for the unseen evaluation can be denoted as $Acc_u$. The H-score is calculated as follows,

$$H_{score} = \frac{2 * Acc_u * Acc_k}{Acc_u + Acc_k}. \tag{6}$$

OSCR is a combination of the accuracy and the AUROC via a moving threshold to measure the quality of the confidence score prediction for the OSDG task. Different from the AUROC, OSDG only calculates the samples that are correctly predicted using such moving threshold, which is a combination of the calculation manner from the H-score and AUROC.

## B  More Ablations of the Proposed Method

### B.1  Ablation of the RBE

We deliver the ablation results of schedulers with different loss function components on the Digits-DG dataset with an open-set ratio of 6:4 for close-set accuracy, H-score, and OSCR of both the conventional and binary classification heads under different dataset partitions (Mnist, Mnist-m, SVHN, SYN). The experiments are conducted by removing the whole $\mathcal{L}_{RBE}$ component or removing only the regularization term $\mathcal{R}_{RB}$ from the EBiL-HaDS method. The results show that EBiL-HaDS performs better than the other two variations on almost all the dataset partitions except the H-score of Mnist-m, as shown in Table 9, demonstrating the effect of our $\mathcal{L}_{RBE}$ and the importance of the regularization term $\mathcal{R}_{RB}$ in the whole $\mathcal{L}_{RBE}$ component.

Comparisons between EBiL-HaDS and the method without the $\mathcal{L}_{RBE}$ illustrate the whole improvement of our deep evidential learning component with regularization term $\mathcal{R}_{RB}$. EBiL-HaDS has in average $2.04\%$, $3.31\%$, $5.07\%$ performance increase of the conventional classification head and $2.04\%$, $3.92\%$, $3.89\%$ increase of the binary classification head for close-set accuracy, H-score, and OSCR, respectively. Differences between the models without $\mathcal{L}_{RBE}$ and without $\mathcal{R}_{RB}$ demonstrate the effect of our deep evidential learning itself. The performance with deep evidential learning improves $0.66\%$, $0.43\%$, $2.47\%$ with the conventional classification head and $0.66\%$, $0.73\%$, $0.83\%$ with the binary classification head for close-set accuracy, H-score, and OSCR in average. On the other hand, EBiL-HaDS has superior results than the model without regularization term $\mathcal{R}_{RB}$ by $1.38\%$, $2.88\%$, $2.60\%$ and $1.38\%$, $3.19\%$, $3.06\%$ for close-set accuracy, H-score, OSCR with the

conventional and binary classification heads averagely. These experiments showcase the significance of regularization term $\mathcal{R}_{RB}$, which contributes more than $50\%$ performance improvement in the whole $\mathcal{L}_{RBE}$ component on all 3 metrics with either conventional or binary classification head.

Table 9: Comparison with different domain schedulers on Digits-DG with open-set ratio 6:4 using ConvNet [66] (Best in **Bold**).

| Method | MNIST | | | MNIST-M | | | SVHN | | | SYN | | | Avg | | |
|---|---|---|---|---|---|---|---|---|---|---|---|---|---|---|---|
| | Acc | H-score | OSCR | Acc | H-score | OSCR | Acc | H-score | OSCR | Acc | H-score | OSCR | Acc | H-score | OSCR |
| w/o RBE (cls) | 99.14 | 79.47 | 95.14 | 72.06 | 59.19 | 56.76 | 77.61 | 56.88 | 58.37 | 91.11 | 72.28 | 72.78 | 84.98 | 66.96 | 70.76 |
| w/o RBE (bcls) | 99.14 | 81.03 | 96.03 | 72.06 | **60.89** | 57.60 | 77.61 | 59.21 | 59.32 | 91.11 | 73.53 | 73.96 | 84.98 | 68.67 | 71.98 |
| w/o RB (cls) | 99.19 | 82.72 | 94.67 | 73.89 | 56.47 | 60.78 | 78.17 | 56.54 | 59.54 | 91.31 | 73.81 | 77.92 | 85.64 | 67.39 | 73.23 |
| w/o RB (bcls) | 99.19 | 84.89 | 96.13 | 73.89 | 60.03 | 58.73 | 78.17 | 61.49 | 60.20 | 91.31 | 71.17 | 76.17 | 85.64 | 69.40 | 72.81 |
| EBiL-HaDS-cls | **99.50** | 87.40 | 97.49 | **74.28** | 56.58 | **60.86** | **80.33** | 61.27 | 62.84 | **93.97** | **75.82** | 78.14 | **87.02** | 70.27 | 75.83 |
| EBiL-HaDS-bcls | **99.50** | **91.63** | **97.58** | **74.28** | 60.72 | 59.39 | **80.33** | **62.23** | **63.88** | **93.97** | 75.77 | **79.28** | **87.02** | **72.59** | **75.87** |

## B.2 Ablation of the Hyperparameters

We visualize the impact of various hyperparameter configurations on the DigitsDG dataset for close-set accuracy, as well as the OSCR of both the conventional and binary classification heads under different dataset partitions (*Mnist*, *Mnist-m*, *SVHN*, *SYN* as unseen domains), as illustrated in Figure 7, Figure 8, and Figure 9. Figure 7 presents an ablation study focusing on the $\sigma$ influences, and demonstrates that varying $\sigma$ from $2e^{-1}$ to $2e^{-6}$ slightly impacts model performance, with certain settings yielding optimal results. For instance, lower values of $\sigma$ generally enhance the OSCR of binary classification head, particularly in more complex datasets like *SVHN*, suggesting that $\sigma$ is important for achieving a balance between robustness and accuracy in domain generalization scenarios.

In Figure 8, adjustments to the loss weight of $L_{REG}$ ranging from $1e^{-3}$ to $1e^{-6}$ are analyzed, which is evident that the loss weight of $L_{REG}$ with $1e^{-4}$ yields the most optimal results across various metrics. Conversely, excessively lower loss weight of $L_{REG}$ appears to have a detrimental effect, potentially leading to overfitting or diminishing the model's ability to generalize effectively across different domains. Besides, Figure 9 explores the ablation of the loss weight of $L_{RBE}$ via adjusting $L_{RBE}$ from $5e^{-1}$ to $5e^{-4}$. It shows a direct effect on both accuracy and OSCR, with lower weights generally improving performance, particularly for challenging datasets like *SVHN*, which suggests that $L_{RBE}$ plays a critical role in the evidential learning framework, significantly influencing the model's ability in open-set conditions under unseen domains. Tuning of hyperparameters $\sigma$, the loss weights of $L_{REG}$ and $L_{RBE}$ are significant for optimizing the performance of domain generalization models. Optimal settings remarkably enhance both the accuracy of meta-training and the OSCR of the classification heads, particularly under complex and challenging dataset conditions like *SVHN* as the unseen domain.

## B.3 Ablation of the Rebiased Layers

In this subsection, we provide the ablation of the layer number used for the rebiased operation as shown in Table 10. In this ablation, 1-1 layer, 2-1 layer, and 2-2 layer indicate that the $R_{\theta_1}$ and $R_{\theta_2}$ are both constructed by 1 convolutional layer, constructed by 1 and 2 convolutional layers, and both constructed by 2 convolutional layers, with unified kernel size 3. We first observe that all of the ablation experiments outperform the baseline MEDIC in terms of the averaged OSDG performance, indicating that with rebiased setting our method can achieve overall OSDG performance improvements regardless of the layer constructions. Delving deeper into the ablation comparison, we find that different convolutional layers to construct the rebiased head can achieve the best performance. This 2-1 layer setting is thereby adopted in other experiments.

## B.4 Comparison with the Self-Generated Reliability-Based Domain Scheduler and Easy-to-Hard Domain Scheduler

We deliver the comparison among the predefined domain scheduler [54], the self-generated reliability-based domain scheduler (denoted as SDGS), the easier domain scheduler (denoted as EDS), and our domain scheduler in Table 11. The self-generated reliability-based domain scheduler indicates that we do not rely on the follower network to achieve the reliability assessment while the confidence

Table 10: Ablation of the rebiased layer number on Digits-DG with open-set ratio 6:4 using ConvNet [66] (Best in **Bold**).

| Method | MNIST | | | MNIST-M | | | SVHN | | | SYN | | | Avg | | |
|---|---|---|---|---|---|---|---|---|---|---|---|---|---|---|---|
| | Acc | H-score | OSCR | Acc | H-score | OSCR | Acc | H-score | OSCR | Acc | H-score | OSCR | Acc | H-score | OSCR |
| MEDIC (cls) | 97.89 | 67.37 | 96.17 | 71.14 | 48.44 | 55.37 | 76.00 | 51.20 | 55.58 | 88.11 | 64.90 | 73.62 | 83.28 | 57.98 | 70.19 |
| MEDIC (bcls) | 97.89 | 83.20 | 95.81 | 71.14 | **60.98** | 58.28 | 76.00 | 58.77 | 57.60 | 88.11 | 62.24 | 72.91 | 83.28 | 66.30 | 71.15 |
| 1-1 layer (cls) | 99.11 | 87.30 | 94.12 | 72.86 | 56.50 | 59.29 | 78.64 | 61.69 | 61.45 | 92.58 | 74.30 | 78.12 | 85.80 | 69.97 | 73.25 |
| 1-1 layer (bcls) | 99.11 | 91.35 | 96.92 | 72.86 | 60.20 | **61.10** | 78.64 | 62.16 | 60.67 | 92.58 | 75.05 | **79.78** | 85.80 | 72.19 | 74.62 |
| 2-1 layer (cls) | **99.50** | 87.40 | 97.49 | **74.28** | 56.58 | 60.86 | **80.33** | 61.27 | 62.84 | 93.97 | **75.82** | 78.14 | **87.02** | 70.27 | 75.83 |
| 2-1 layer (bcls) | **99.50** | **91.63** | **97.58** | **74.28** | 60.72 | 59.39 | **80.33** | **62.23** | **63.88** | 93.97 | 75.77 | 79.28 | **87.02** | **72.59** | **75.87** |
| 2-2 layers (cls) | 99.08 | 87.02 | 96.61 | 73.32 | 56.57 | 59.33 | 79.14 | 61.97 | 61.03 | **94.00** | 74.54 | 79.13 | 86.39 | 70.03 | 74.03 |
| 2-2 layers (bcls) | 99.08 | 90.12 | 96.79 | 73.32 | 60.11 | 59.59 | 79.14 | 59.58 | 60.04 | **94.00** | 72.74 | 77.48 | 86.39 | 70.64 | 73.48 |

score from the main network is utilized for the domain scheduling. This comparison is designed to showcase the importance of the follower network used for the domain reliability assessment. The easier domain scheduler indicates that we use the domain with the highest reliability to accomplish the data partition during the meta-learning.

Through using the follower network, we observe that our method outperforms SDGS by 2.19%, 2.81%, and 3.29% of Acc, H-score, and OSCR for binary classification head and 2.19%, 3.42%, and 4.05% of Acc, H-score, and OSCR for conventional classification head. Consistent performance benefits of our method can be observed when we compare the results of the EDS with ours, indicating the importance of using the hardest domain scheduler in the OSDG task when meta-learning is involved. Compared with the predefined domain scheduler from MEDIC, SDGS and EDS outperform it obviously, indicating the importance of using an adaptive domain scheduler during the meta-learning procedure for the OSDG task.

Table 11: Comparison of self-generated reliability-based domain scheduler, the Easy2Hard scheduler, and ours on Digits-DG with open-set ratio 6:4 using ConvNet [66] (Best in **Bold**).

| Method | MNIST | | | MNIST-M | | | SVHN | | | SYN | | | Avg | | |
|---|---|---|---|---|---|---|---|---|---|---|---|---|---|---|---|
| | Acc | H-score | OSCR | Acc | H-score | OSCR | Acc | H-score | OSCR | Acc | H-score | OSCR | Acc | H-score | OSCR |
| MEDIC (cls) | 97.89 | 67.37 | 96.17 | 71.14 | 48.44 | 55.37 | 76.00 | 51.20 | 55.58 | 88.11 | 64.90 | 73.62 | 83.28 | 57.98 | 70.19 |
| MEDIC (bcls) | 97.89 | 83.20 | 95.81 | 71.14 | **60.98** | 58.28 | 76.00 | 58.77 | 57.60 | 88.11 | 62.24 | 72.91 | 83.28 | 66.30 | 71.15 |
| SDGS (cls) | 98.97 | 85.99 | 95.67 | 71.17 | 50.86 | 56.06 | 76.72 | 57.55 | 57.78 | 92.47 | 72.98 | 77.61 | 84.83 | 66.85 | 71.78 |
| SDGS (bcls) | 98.97 | 88.79 | 96.62 | 71.17 | 57.01 | 54.18 | 76.72 | 57.89 | 59.06 | 92.47 | 75.44 | 80.46 | 84.83 | 69.78 | 72.58 |
| EDS (cls) | 98.89 | 76.21 | 95.40 | 72.17 | 53.81 | 57.46 | 76.86 | 58.07 | 58.95 | 90.72 | 69.40 | 76.45 | 84.66 | 64.37 | 72.07 |
| EDS (bcls) | 98.89 | 88.17 | 96.46 | 72.17 | 59.15 | 56.09 | 76.86 | 59.61 | 57.97 | 90.72 | 69.98 | 76.58 | 84.66 | 69.23 | 71.78 |
| Ours (cls) | **99.50** | 87.40 | 97.49 | **74.28** | 56.58 | **60.86** | **80.33** | 61.27 | 62.84 | **93.97** | **75.82** | 78.14 | **87.02** | 70.27 | 75.83 |
| Ours (bcls) | **99.50** | **91.63** | **97.58** | **74.28** | 60.72 | 59.39 | **80.33** | **62.23** | **63.88** | **93.97** | 75.77 | **79.28** | **87.02** | **72.59** | **75.87** |

## B.5 Comparison with Other Curriculum Learning Approaches

We implement two recent curriculum learning approaches, *i.e.*, the training paradigms proposed by Wang *et al.* [56] and Abbe *et al.* [1] into the OSDG task, where the comparison of the experimental results are delivered in Table 12. We first observe that using different training paradigms can benefit OSDG. Compared with the MEDIC baseline, the approach proposed by Abbe *et al.* [1] achieves performance improvements of 2.44%, 3.49%, and 0.92% and 2.44%, 2.04%, and 1.09% of Acc, H-score, and OSCR for the conventional classification head and the binary classification head. The approach proposed by Wang *et al.* [56] also delivers promising comparable results with the MEDIC baseline. Furthermore, since our approach is specifically designed for the OSDG task, our approach achieves performance improvements of 1.30%, 8.80%, and 4.72% and 1.30%, 4.25%, and 3.63% in Acc, H-score, and OSCR for the conventional classification head and the binary classification head compared with the approach proposed by Abbe *et al.* [1], illustrating the superior performance of our domain scheduler based data partition in the OSDG task.

## C  More Details of the Open-Set Ratios and Splits Ablations

We visualize the impact of different strategies for splitting known and unknown classes on the DigitsDG dataset, specifically examining the effects on close-set accuracy and OSCR of both conventional and binary classification heads. The analysis covers various dataset partitions, including *Mnist*, *Mnist-m*, *SYN*, and *SVHN*, as depicted in Figure 4 and Figure 5.  Figure 4 presents an

Table 12: Comparison with other curriculum learning methods on Digits-DG with open-set ratio 6:4 using ConvNet [66] (Best in **Bold**).

| Method | MNIST | | | MNIST-M | | | SVHN | | | SYN | | | Avg | | |
|---|---|---|---|---|---|---|---|---|---|---|---|---|---|---|---|
| | Acc | H-score | OSCR | Acc | H-score | OSCR | Acc | H-score | OSCR | Acc | H-score | OSCR | Acc | H-score | OSCR |
| MEDIC (cls) | 97.89 | 67.37 | 96.17 | 71.14 | 48.44 | 55.37 | 76.00 | 51.20 | 55.58 | 88.11 | 64.90 | 73.62 | 83.28 | 57.98 | 70.19 |
| MEDIC (bcls) | 97.89 | 83.20 | 95.81 | 71.14 | **60.98** | 58.28 | 76.00 | 58.77 | 57.60 | 88.11 | 62.24 | 72.91 | 83.28 | 66.30 | 71.15 |
| Wang *et al.* [56] (cls) | 98.92 | 87.80 | 94.34 | 72.22 | 48.27 | 57.99 | 78.14 | 52.87 | 59.92 | 90.03 | 62.48 | 69.35 | 84.83 | 62.86 | 70.40 |
| Wang *et al.* [56] (bcls) | 98.92 | 89.42 | 94.98 | 72.22 | 60.13 | 58.22 | 78.14 | 59.20 | 59.46 | 90.03 | 66.43 | 70.67 | 84.83 | 68.78 | 70.83 |
| Abbe *et al.* [1] (cls) | 99.22 | 78.06 | 94.79 | 71.92 | 50.16 | 55.56 | 79.81 | 52.80 | 59.95 | 91.92 | 64.87 | 74.13 | 85.72 | 61.47 | 71.11 |
| Abbe *et al.* [1] (bcls) | 99.22 | 88.37 | 95.53 | 71.92 | 60.06 | 57.96 | 79.81 | 59.55 | 60.32 | 91.92 | 65.37 | 75.14 | 85.72 | 68.34 | 72.24 |
| Ours (cls) | **99.50** | 87.40 | 97.49 | **74.28** | 56.58 | **60.86** | **80.33** | 61.27 | 62.84 | **93.97** | **75.82** | 78.14 | **87.02** | 70.27 | 75.83 |
| Ours (bcls) | **99.50** | **91.63** | **97.58** | **74.28** | 60.72 | **59.39** | **80.33** | 62.23 | **63.88** | **93.97** | 75.77 | **79.28** | **87.02** | **72.59** | **75.87** |

ablation study focusing on the selection of unknown classes with a 6:4 ratio. Despite the suboptimal dataset partitioning leading to declines in both close-set accuracy and OSCR, the superiority of our method remains largely unaffected by the choice of unknown classes. It consistently achieves the highest close-set accuracy and OSCR across most domains within the DigitsDG dataset, while also maintaining competitive performance in the remaining domains. This demonstrates the model's proficiency in distinguishing and recognizing known classes within the training set, as well as its capability of managing unseen classes, thereby highlighting its robustness in open-set environments. Furthermore, Figure 5 illustrates an ablation study examining various open-set ratios (7:3 and 8:2). By analyzing the impact of various open-set ratios in this ablation, we show that our method effectively mitigates saturation in close-set accuracy, maintaining robust generalization capabilities of OSDG. However, the imbalance between known and unknown samples diminishes the model's ability to differentiate unknown classes, resulting in a general decrease in OSCR. Despite this challenge, our method consistently achieves the highest OSCR across different open-set ratios compared with the baselines. Notably, in the complex and challenging *SVHN* domain, our OSCR exceeds the baseline MEDIC-bcls by $5.08\%$ at the 8:2 ratio. This finding underscores our model's exceptional performance in detecting unknown classes and accurately classifying known ones.

## D   Analysis of the Performances during Training

We visualize the accuracy changes during training every 100 epochs for the meta-learning process, reporting both validation accuracy and test accuracy, as shown in Figure 6. Implementing our domain scheduler introduces significant improvements in the model's training dynamics. Notably, the validation accuracy curve appears smoother compared to the MEDIC baseline, as illustrated in Figure 6a and Figure 6c. At the early stages of meta-learning for OSDG, our domain scheduler demonstrates superior initial training performance, with test set accuracy surpassing that of MEDIC [54]. These findings suggest that customizing the training schedule to match distinct domain partitions within the data can substantially enhance both training efficiency and overall performance. Our approach capitalizes on specialized domain knowledge, allowing the training algorithm to better adapt to varying data characteristics, ultimately optimizing the model's performance outcomes.

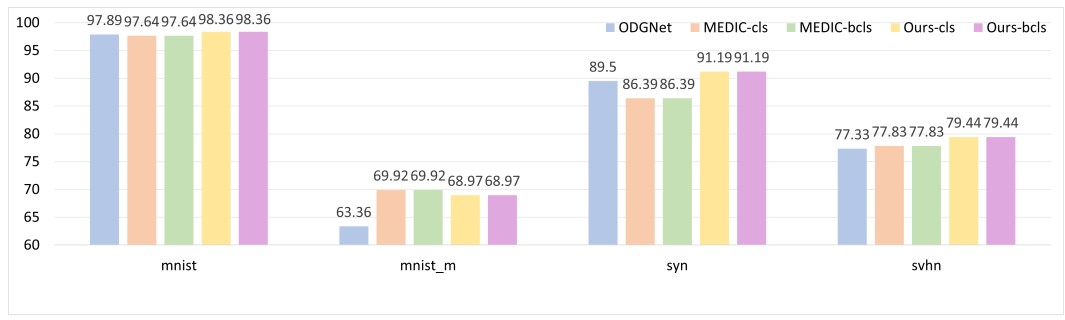

(a) Close-set accuracy on DigitsDG when we choose 0, 1, 2, 3 as unknown classes.

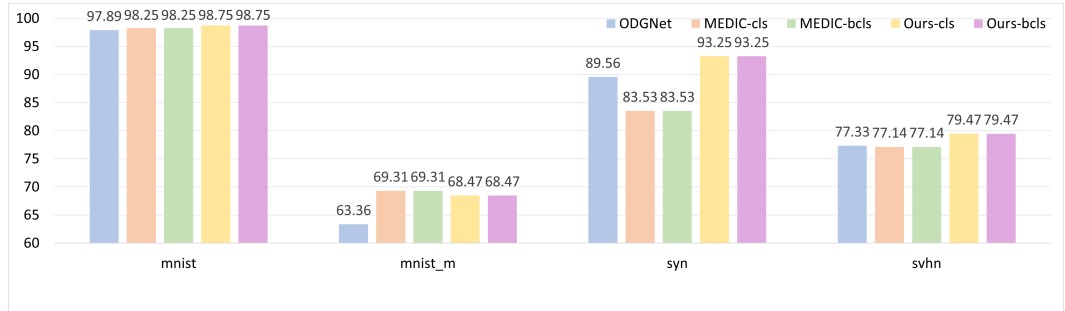

(b) Close-set accuracy on DigitsDG when we choose 2, 3, 4, 5 as unknown classes.

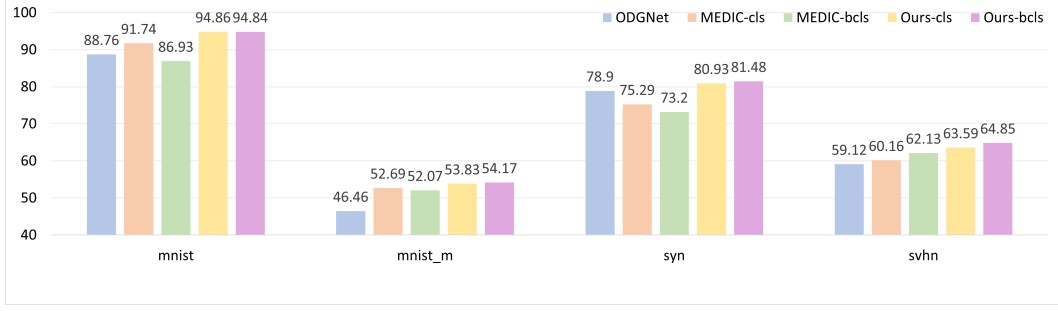

(c) OSCR on DigitsDG when we choose 0, 1, 2, 3 as unknown classes.

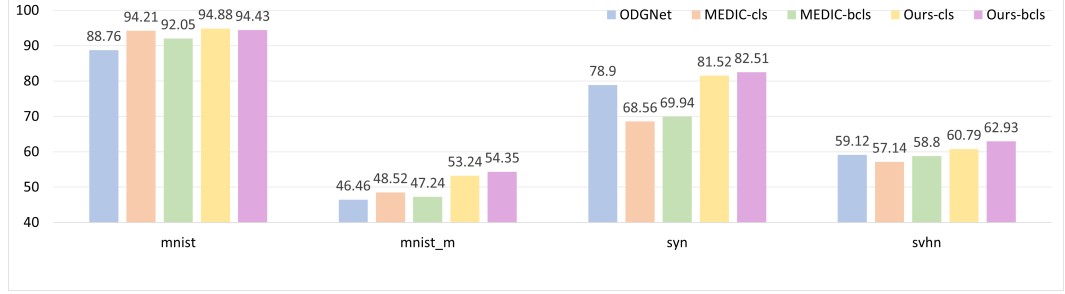

(d) OSCR on DigitsDG when we choose 2, 3, 4, 5 as unknown classes.

Figure 4: Experimental details for the ablation of different splits on 6:4 ratio on DigitsDG dataset. (Supplementary figure)

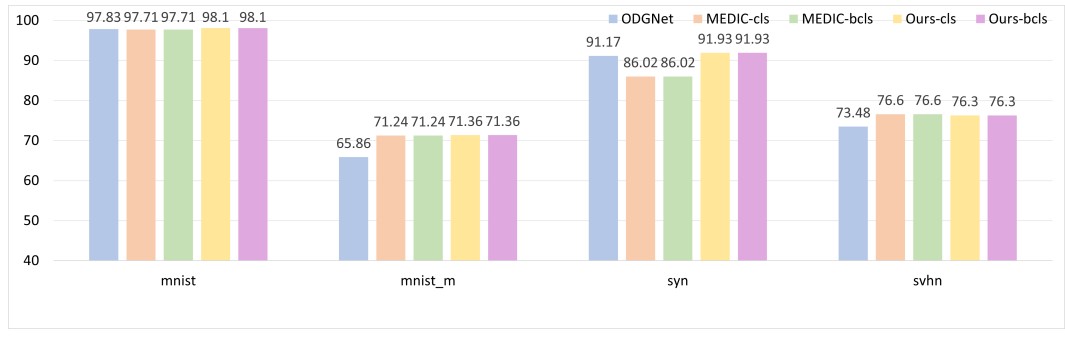

(a) Close-set accuracy on DigitsDG when we choose 7, 8, 9 as unknown classes.

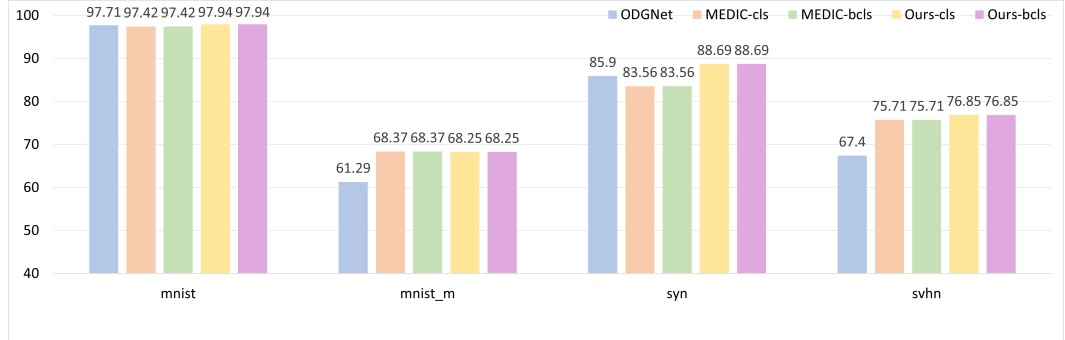

(b) Close-set accuracy on DigitsDG when we choose 8, 9 as unknown classes.

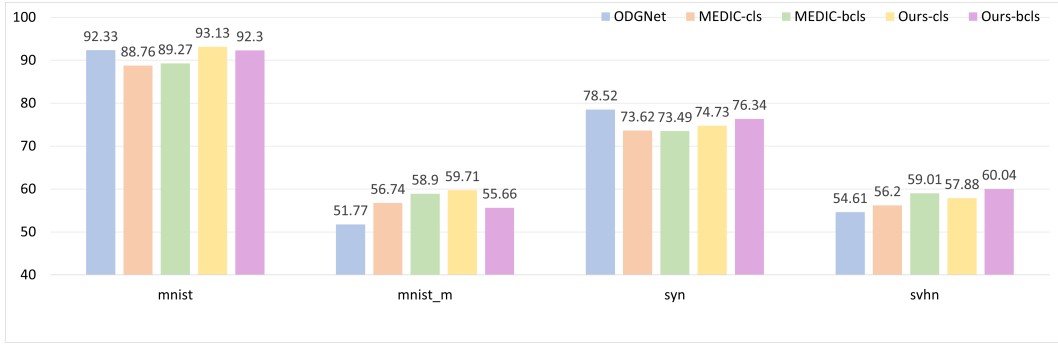

(c) OSCR on DigitsDG when we choose 7, 8, 9 as unknown classes.

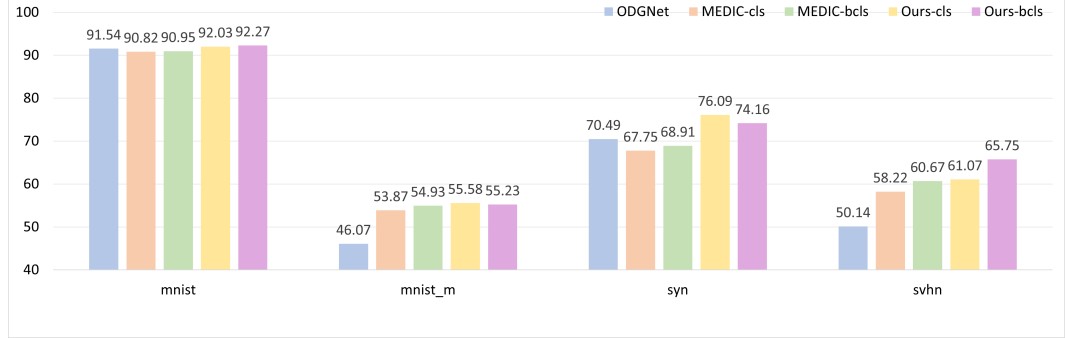

(d) OSCR on DigitsDG when we choose 8, 9 as unknown classes.

Figure 5: Experimental details for the ablation of different open-set ratios on DigitsDG dataset. (Supplementary figure)

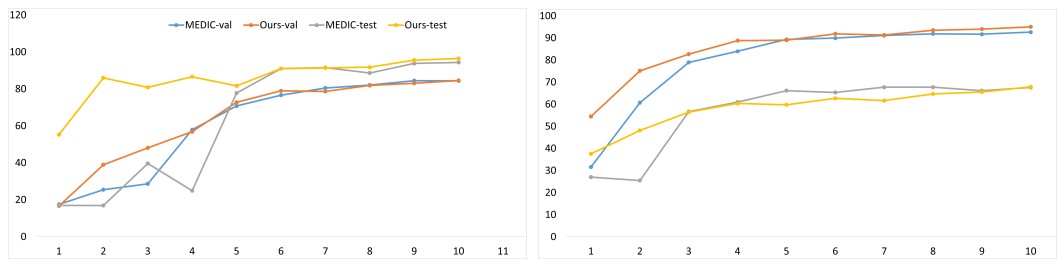

(a) Val and test accuracy on MnistDG, where Mnist is chosen as the unseen domain.

(b) Val and test accuracy on MnistDG, where Mnist-m is chosen as the unseen domain.

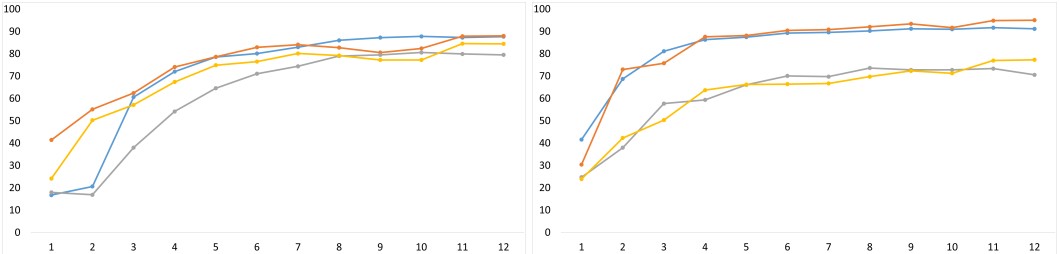

(c) Val and test accuracy on MnistDG, where SYN is chosen as the unseen domain.

(d) Val and test accuracy on MnistDG, where SVHN is chosen as the unseen domain.

Figure 6: Val and test accuracy on MnistDG, where the validation accuracy of MEDIC and our approach are indicated by lines in blue and orange colors, and the test accuracy of MEDIC and our approach are indicated by lines in gray and yellow colors. The horizontal axis indicates the evaluation step with stepsize 100 during the meta-learning procedure.

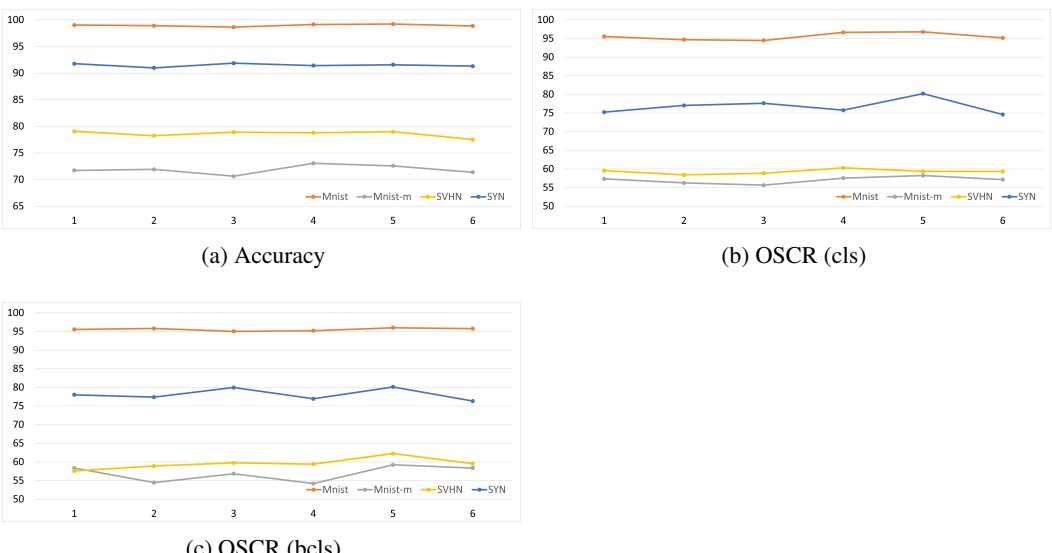

(a) Accuracy

(b) OSCR (cls)

(c) OSCR (bcls)

Figure 7: Ablation for the $\sigma$, where the horizontal axis indicates the ablation cases. Case 1, 2, 3, 4, 5, and 6 indicate $2e^{-1}$, $2e^{-2}$, $2e^{-3}$, $2e^{-4}$, $2e^{-5}$ and $2e^{-6}$. The experiments are conducted on DigitsDG using a ConvNet architecture.

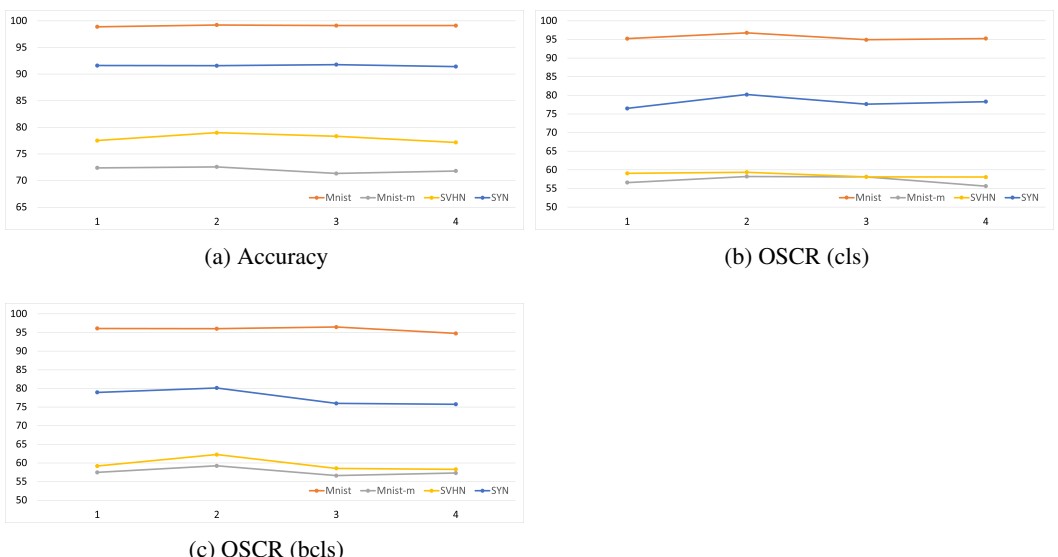

(a) Accuracy

(b) OSCR (cls)

(c) OSCR (bcls)

Figure 8: Ablation for the loss weight of the $L_{REG}$, where the horizontal axis indicates the ablation cases. Case 1, 2, 3, and 4 indicate $1e^{-3}$, $1e^{-4}$, $1e^{-5}$, and $1e^{-6}$. The experiments are conducted on DigitsDG using a ConvNet architecture.

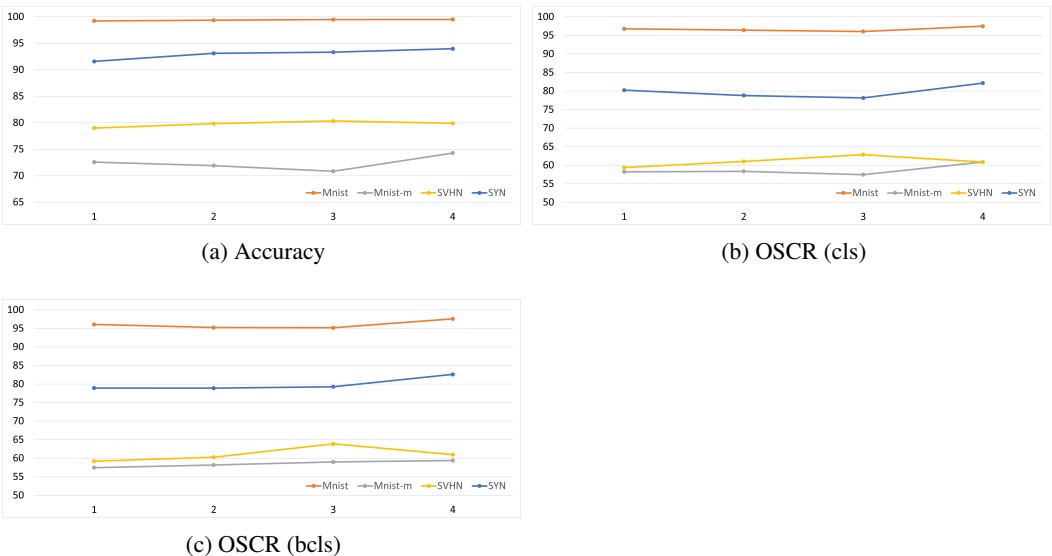

(a) Accuracy

(b) OSCR (cls)

(c) OSCR (bcls)

Figure 9: Ablation for the loss weight of the $L_{RBE}$, where the horizontal axis indicates the ablation cases. Case 1, 2, 3, and 4 indicate $5e^{-1}$, $5e^{-2}$, $5e^{-3}$, and $5e^{-4}$. The experiments are conducted on DigitsDG using a ConvNet architecture.

