# OpenReview forum: "Advancing Open-Set Domain Generalization Using Evidential Bi-Level Hardest Domain Scheduler"
_NeurIPS.cc/2024/Conference — NeurIPS 2024 poster_

### Official Review · Reviewer_tsHR · 2024-07-07

**Soundness:** 2
**Presentation:** 3
**Contribution:** 2
**Rating:** 5
**Confidence:** 3

**Summary:**

In this paper, the authors identify a new problem in open-set domain generalization, proposing that dynamically adapting the domain scheduler used for data partitioning based on specific criteria could lead to a more targeted training strategy and improved outcomes. They introduce a novel training strategy named the Evidential Bi-Level Hardest Domain Scheduler (EBiL-HaDS), which combines max rebiased discrepancy evidential learning with meta-learning to create an adaptive domain scheduler.

**Strengths:**

1. The authors focus on a new and interesting problem that dynamically adapt the domain scheduler used for data partition may improve open-set domain generalization performance.
2. Extensive experiments verify the efficiency of the proposed adaptive domain scheduler.

**Weaknesses:**

1. The motivation for considering the training order of domains is unclear. I believe that if there were temporal correlation between domains, considering the training order would be reasonable. However, in this research setting and the datasets used in the experiments, there is no temporal correlation. Therefore, why should the training order be considered?

2. In my opinion, this paper primarily makes some improvements to the MEDIC model. However, compared to MEDIC, the proposed method does not show significant performance improvement on some tasks.

3. Dynamically adapting the domain scheduler used for data partitioning based on certain criteria can be seen as a trick rather than a major contribution.

4. The key idea of EBiL-HaDS is to quantify domain reliability. I suggest that the authors provide a more detailed definition of domain reliability and demonstrate that domain reliability is existing in real-world scenarios with a significant impact on generalization.

5. What is the motivation for applying max rebiased discrepancy evidential learning to achieve more reliable confidence acquisition? Are there any particular advantages to this approach? The authors should compare this approach with other similar techniques.

6. There are still some grammatical mistakes and typos in this paper. Please proofread the paper carefully.

**Questions:**

Discussed in the weaknesses.

**Limitations:**

Yes.

---

> ### Author Rebuttal · Authors · 2024-08-06
>
> We appreciate the review effort from Reviewer tsHR and provide the point-to-point responses as follows. All the explanations will be included into our final paper.
>
>
> **W1**: Training order scheduler is very important in deep learning for some specific techniques, e.g., meta learning and curriculum learning. In curriculum learning, researchers propose the instance scheduler to decide the next training samples to pursue a better optimization of the performance of the deep learning model, where the samples do not contain temporal correlation, as demonstrated in works, [a] and [b].
> Meta-learning aims at reserving different training tasks for model optimization, where the training order can guide the model to different optimization situations. Choosing different tasks/samples for training at different stages can result in obvious performance differences in the models.
> MEDIC and MLDG propose to use meta learning techniques for OSDG, where meta learning shows strong OSDG capability when the meta training and meta testing are conducted on different source domains. However, both of them rely on sequential source domain order when setting up the meta tasks. Our experiments turn out that sequential domain scheduler is not the optimal way for domain scheduling in the meta learning of the OSDG task. In this work, we for the first time propose an adaptive domain scheduler in a bi-level optimization manner, which shows strong OSDG capability across different backbones and datasets.
>
> We conducted the ablation in Table 6 of our main paper to demonstrate the performance gains brought by our proposed adaptive domain scheduler, when we compare our proposed domain scheduler method with random domain scheduler and sequential domain scheduler.
>
> [a] Yulin Wang, Yang Yue, Rui Lu, Tianjiao Liu, Zhao Zhong, Shiji Song, and Gao Huang. Efficienttrain: Exploring generalized curriculum learning for training visual backbones. In ICCV, 2023
>
> [b] Emmanuel Abbe, Samy Bengio, Aryo Lotfi, and Kevin Rizk. Generalization on the unseen, logic reasoning and degree curriculum. In ICML, 2023
>
>
> **W2**: In our paper we for the first time propose an adaptive evidential bi-level optimized domain scheduler for the meta learning used in the OSDG task, where the existing works, i.e., MLDG and MEDIC, rely on sequential domain scheduler. Our approaches achieve consistent state-of-the-art performances across 3 different DG benchmarks where 4 different domain generalization settings are explored. In the response to the Reviewer QWTs, we provide more ablations on larger backbones, i.e., ResNet152 and ViT-base, where obvious OSCR gains are observed. All of these experimental results showcase the strong capability of our proposed method towards open-set domain generalization challenge.
>
> In Table 9 of the appendix, we conducted ablation experiments to justify the effectiveness of our model with more variants of domain scheduler for meta learning on OSDG task.
>
>
> **W3**: We clarify that the adaptive domain scheduler is our major contribution, which is proposed by us in this work to estimate the uncertainty level of different source domains during the meta-learning procedure using an evidentially bi-level optimized follower network. Unlike curriculum learning (e.g., [a] and [b]), which focuses on scheduling the training order of different samples or batches, we for the first time propose the adaptive domain scheduler to achieve adaptive reservation of different source domains for meta-learning approaches within the realm of OSDG task. We demonstrate that the proposed domain scheduler is much more powerful compared with other curriculum learning strategies as proved by the experimental results in Table 10 in the appendix.
>
>
> **W4**:  Domain reliability is based on the statistic mean of the uncertainty estimation of the samples from one specific domain, estimated by the follower network. The prediction generated by the bi-level optimized follower network is used as uncertainty quantification of the corresponding input sample, since this follower network is optimized by the evidentially learned uncertainty score from the main network.
> From our experiments, we observe that the  data partition based on the most unreliable domain, where the uncertainty estimation of this source domain from follower network is the highest, benefit the OSDG performances. In Table 9 of the appendix, we provide further ablation between the domain scheduler achieved by bi-level optimized follower and the uncertainty estimation of the main network. Our method shows better performances.
>
>
> **W5**: Evidential learning is a well-established technique to pursue better uncertainty estimation of the deep learning methods. However, as mentioned in related work [c], evidential learning faces with challenge of overfitting. In this work, we propose to use max rebiased discrepancy to force the two rebiased layers to learn different cues to overcome this challenge, where the max rebiased discrepancy serves as additional regularization for the evidential learning of the main network. After the supervision of the max rebiased discrepancy evidential loss, the main network can supervise the follower network well by providing more reliable estimated uncertainty.
>
> In Table 7 in the appendix, we conducted experiments to illustrate the benefits brought by the max rebiased discrepancy regularization term, where the discussion of this ablation is demonstrated in Section B.1 in our appendix. Our observation turns out that by using the proposed max rebiased discrepancy regularization, the follower network can be optimized better and more reliable in the domain reliability quantification, thereby the proposed method can achieve higher OSDG performances.
>
> [c] Danruo Deng, Guangyong Chen, Yang Yu, Furui Liu, and Pheng-Ann Heng. Uncertainty estimation by fisher information-based evidential deep learning. In ICML, 2023.
>
> **W6**: The corresponding typos will be changed in our final version, thank you.

---

> > ### Author Response · Authors · 2024-08-12
> > **To Reviewer tsHR**
> >
> > Dear Reviewer tsHR,
> >
> > Thank you very much for your review effort and questions.
> >
> > The authors would like to ask if you have any further questions on the rebuttal reply, since the author reviewer discussion period will end soon.
> >
> >
> > The point-to-point response of each question in the weakness part is provided in the rebuttal response. In case there is anything unclear please feel free to let us know.
> >
> > Thank you.
> >
> > Best,
> >
> > The authors.

---

> > > ### Comment · Reviewer_tsHR · 2024-08-12
> > >
> > > Thanks for the authors' detailed response and additional experiments. I will increase the score to 5.

---

> > > > ### Author Response · Authors · 2024-08-12
> > > > **To Reviewer tsHR**
> > > >
> > > > Dear Reviewer tsHR,
> > > >
> > > > Thank you for your fast feedback. The authors are pleased with your decision and will conduct all your suggested modifications in the final version.
> > > >
> > > > Best,
> > > >
> > > > The authors.

---

### Official Review · Reviewer_euSj · 2024-07-10

**Soundness:** 3
**Presentation:** 2
**Contribution:** 3
**Rating:** 5
**Confidence:** 4

**Summary:**

The paper addresses the challenges of Open-Set Domain Generalization (OSDG) and introduces the Evidential Bi-Level Hardest Domain Scheduler (EBiL-HaDS), which adaptively sequences domains based on their reliability, assessed through a follower network. The authors verify EBiL-HaDS on three benchmarks: PACS, DigitsDG, and OfficeHome, demonstrating substantial performance improvements in OSDG.

**Strengths:**

1. EBiL-HaDS dynamically adjusts the domain training sequence, which is a novel idea compared to the fixed schedulers used in previous works.
2. Comprehensive experiments and ablations on PACS, DigitsDG, and OfficeHome demonstrate the effectiveness of the proposed approach
3. The paper is very well written and easy to follow

**Weaknesses:**

1. The proposed framework is too complex and hard to reproduce. The source code is also not provided.
2. The authors only use CNN backbones. More ablations on ViT backbone should be added, as it demonstrates strong domain generalization performances compared with CNN [A, B]

[A] Li et al., Sparse Mixture-of-Experts are Domain Generalizable Learners, ICLR 2023

[B] Zheng et al., Prompt Vision Transformer for Domain Generalization, arxiv 2022

**Questions:**

What is the performance under ViT architecture?

**Limitations:**

The authors discuss the limitation of potential leading to misclassification and biased content, potentially causing false predictions.

---

> ### Author Rebuttal · Authors · 2024-08-06
>
> Thank you for recognizing the motivation and contribution of the proposed method.
>
> **W1:** The anonymous link to the source code has been submitted in the official comment to AC which can be helpful for the results reproduction. The source code will be released publicly in the final version.
>
>
> **W2:** We have conducted additional 24 experiments on the ViT-base model according to your suggestion in Table A of this response. The results can be found below, where our proposed method is compared with the MEDIC and other challenging baselines. We can observe that our proposed method achieves state-of-the-art performances when we use ViT-base model as the feature extraction backbone.
>
> Compared with the MEDIC method, our EBil-HaDS achieves 1.69% and 2.05% performance improvements in terms of OSCR for standard classification head and binary classification head, demonstrating the capability of the proposed adaptive domain scheduler on the generalization to different feature extraction backbones. The corresponding experiments and discussion will be added to the final version of our paper.
>
>
> **Table A: Ablation for ViT base backbone**
> | Method          | Photo (P)  Acc | Photo (P) H-score | Photo (P) OSCR | Art (A) Acc | Art (A) H-score | Art (A) OSCR | Cartoon (C) Acc | Cartoon (C) H-score | Cartoon (C) OSCR | Sketch (S) Acc | Sketch (S) H-score | Sketch (S) OSCR | Avg Acc | Avg H-score | Avg OSCR |
> |-----------------------|---------------|-------------------|---------------|-------------|-----------------|--------------|-----------------|---------------------|-----------------|----------------|---------------------|-----------------|----------|--------------|-----------|
> | ARPL | 99.19| 95.31| 98.61| 90.49 | 85.46 | 88.59| 81.88| 72.17| 73.34| 63.01| 29.33| 50.59| 83.64| 70.57| 77.78|
> | MLDG| 99.19| 95.40| 98.88| 91.87| 82.46| 89.47| 80.56| 69.62| 74.19| 61.66| 40.79| 43.88| 83.32 | 72.07 | 76.61|
> | SWAD| 98.55| 93.19| 97.62| 90.81| 81.34| 88.52| 83.24 | 73.03| 76.59| 57.89| 35.83| 41.68 | 82.62 | 70.85       | 76.10     |
> |ODG-Net|97.58	|96.24	|95.23	|90.49	|83.32	|87.90	|82.36	|68.66	|75.80	|62.59	|43.59	|50.22	|83.26	|72.95	|77.29	|
> |MEDIC-cls	|99.03	|95.33	|98.22	|92.06	|83.27	|87.46	|85.62	|69.79	|75.37	|68.40	|41.95	|56.56	|86.28	|72.59	|79.40	|
> |MEDIC-bcls|99.03	|96.04	|97.55	|92.06	|82.68	|87.73	|85.62	|69.15	|76.80	|68.40	|39.60	|55.92	|86.28	|71.87	|79.50	|
> |ours-cls|**99.52**	|**97.30**|99.11|**94.68**|86.10	|92.10	|**89.22**	|**74.31**	|77.76	|**69.49**	|44.34	|55.37	|**88.23**|75.53	|81.09	|
> |ours-bcls|**99.52**	|96.91	|**99.18**	|**94.68**|**88.31**	|**92.28**	|**89.22**	|73.91	|**77.95**	|**69.49**	|**48.09**	|**56.78**	|**88.23**|**76.81**|**81.55**|

---

> ### Author Response · Authors · 2024-08-12
> **To Reviewer euSj**
>
> Dear Reviewer euSj,
>
>
> Thank you very much for your comments and your contribution to the review procedure.
>
> The authors would like to ask if you have any question on the rebuttal, since the discussion period between the authors and the reviewers will end soon.
>
> We provided the source code link and the additional 24 ablation experiments on ViT-Base according to your instruction in the rebuttal. Please let us know if you have any further questions.
>
> Thank you.
>
> Best,
>
> The authors.

---

> > ### Comment · Reviewer_euSj · 2024-08-12
> >
> > Thank the authors for their rebuttal. Most of my concerns have been well-addressed and I thus increased my score to 5.

---

> > > ### Author Response · Authors · 2024-08-12
> > > **To Reviewer euSj**
> > >
> > > Dear Reviewer euSj,
> > >
> > > Thank you very much for your response and decision, the authors are very glad to hear your decision. We will include all the experiments and source code link in our final version.
> > >
> > > Thank you.
> > >
> > > Best,
> > >
> > > The authors

---

### Official Review · Reviewer_QWTs · 2024-07-12

**Soundness:** 2
**Presentation:** 2
**Contribution:** 2
**Rating:** 5
**Confidence:** 3

**Summary:**

In this paper, an observation is proposed, that an adaptive domain scheduler benefits more in OSDG compared with prefixed sequential and random domain schedulers. A follower network is proposed to strategically sequences domains by assessing their reliabilities, which is trained with confidence scores learned in an evidential manner and optimized in a bi-level manner. Experiments show the superior OSDG performance and ability to get more discriminative embeddings for both the seen and unseen categories.

**Strengths:**

1. The paper is well-written.
2. The influence of domain scheduling in the OSDG task remains unexplored. This paper examines the effects of guiding the meta-learning process with an adaptive domain scheduler. A domain reliability measure method is proposed by an follower network.
3. The proposed method conducts comprehensive and discriminative data partitions during training, enhancing generalization to unseen domains.
4. Experiments on OSDG tasks show the superior performance.

**Weaknesses:**

The experiments conducted only on ResNet50 ResNet18 and ConvNet. The number of parameters in those backbone are limited. If the comparison is conducted on models with a large number of parameters, like ResNet152 or ViT-based models, would the advantage in classification performance be leveled off?

**Questions:**

If the comparison is conducted on models with a large number of parameters, like ResNet152 or ViT-based models, would the advantage in classification performance be leveled off?

**Limitations:**

The experiments only conducted on classification tasks.

---

> ### Author Rebuttal · Authors · 2024-08-06
>
> Thank you for recognizing the motivation and contribution of our proposed method. We have conducted an additional 48 experiments using the ResNet152 and ViT base models. The results, detailed below, compare our proposed method with MEDIC and other challenging baselines such as ARPL, MLDG, SWAD, and ODG-Net. From Tables 1 and 2 in our main paper, we observe a significant OSDG performance improvement when increasing the complexity of the feature extraction backbone. However, when comparing Table A (rebuttal) with Table 2 (main paper), some baseline approaches trained with the ResNet152 backbone show performance decay on the major evaluation metric, OSCR.
>
> This observation indicates that most OSDG methods face overfitting issues with larger backbones, which is particularly problematic for recognizing samples from unseen categories in unseen domains. Specifically, the MEDIC-bcls approach shows a 6.48% performance degradation on OSCR when switching from ResNet50 to ResNet152. In contrast, using the proposed BHiL-HaDS for more reasonable task reservation in the meta-learning procedure, BHiL-HaDS-bcls achieves 86.25% OSCR with the ResNet152 backbone, compared to 86.12% OSCR with the ResNet50 backbone. This demonstrates that our adaptive domain scheduler can effectively optimize meta-learning for large complex models by reserving appropriate tasks for model optimization.
>
> **Table A: Ablation for ResNet152 backbone**
> | Method            | Photo (P)  Acc | Photo (P) H-score | Photo (P) OSCR | Art (A) Acc | Art (A) H-score | Art (A) OSCR | Cartoon (C) Acc | Cartoon (C) H-score | Cartoon (C) OSCR | Sketch (S) Acc | Sketch (S) H-score | Sketch (S) OSCR | Avg Acc | Avg H-score | Avg OSCR |
> |-----------------------|---------------|-------------------|---------------|-------------|-----------------|--------------|-----------------|---------------------|-----------------|----------------|---------------------|-----------------|----------|--------------|-----------|
> |ARPL	|94.35|85.45|86.74|89.81|71.27|78.53|83.91	|69.75|72.08|77.53	|52.70|66.68|77.53	|69.81	|76.01	|
> |MLDG	|96.20|91.07|94.64|89.81|77.65|82.19|83.86	|73.66|74.03|82.89	|64.30|72.98|88.19	|76.67	|80.96|
> |SWAD	|95.64|84.82|89.74	|86.30|73.86|75.91|78.49|70.18|68.41|76.92|75.33|63.35|84.34	|76.05	|74.35	|
> |ODG-Net|95.88|89.11|91.85|89.62|80.65|82.48|85.15|70.37|73.66|79.30|77.00|72.22|87.49	|79.28	|80.05	|
> |MEDIC-cls|94.67	|49.54|76.98|89.37|73.26|77.79|86.59|68.49|74.82|85.81|56.14|78.83	|89.11	|61.86|77.11|
> |MEDIC-bcls|94.67|72.88|81.30|89.37|74.92|78.70|86.59|71.46|75.17|85.81|58.80|78.32	|89.11	|69.52	|78.37|
> |ours-cls|**97.90**	|91.66|96.62|**92.06**|81.52|85.43|**87.21**|76.61|78.19|**87.08**|81.13	|80.21|**91.06**|82.73	|85.11|
> |ours-bcls|**97.90**|**94.34**	|**97.39**|**92.06**|**82.00**	|**85.94**	|**87.21**	|**76.62**	|**80.15**	|**87.08**	|**88.57**	|**81.52**	|**91.06**|**85.38**|**86.25**|
>
>
>
> Additional experiments on the ViT-base model (patch size 16 and window size 224) are presented in Table B of this response. We observe that our proposed method consistently delivers performance gains on the transformer architecture. These ablation experiments and discussions will be included in our final paper.
>
>
> **Table B: Ablation for ViT base backbone**
> | Method          | Photo (P)  Acc | Photo (P) H-score | Photo (P) OSCR | Art (A) Acc | Art (A) H-score | Art (A) OSCR | Cartoon (C) Acc | Cartoon (C) H-score | Cartoon (C) OSCR | Sketch (S) Acc | Sketch (S) H-score | Sketch (S) OSCR | Avg Acc | Avg H-score | Avg OSCR |
> |-----------------------|---------------|-------------------|---------------|-------------|-----------------|--------------|-----------------|---------------------|-----------------|----------------|---------------------|-----------------|----------|--------------|-----------|
> | ARPL | 99.19| 95.31| 98.61| 90.49 | 85.46 | 88.59| 81.88| 72.17| 73.34| 63.01| 29.33| 50.59| 83.64| 70.57| 77.78|
> | MLDG| 99.19| 95.40| 98.88| 91.87| 82.46| 89.47| 80.56| 69.62| 74.19| 61.66| 40.79| 43.88| 83.32 | 72.07 | 76.61|
> | SWAD| 98.55| 93.19| 97.62| 90.81| 81.34| 88.52| 83.24 | 73.03| 76.59| 57.89| 35.83| 41.68 | 82.62 | 70.85       | 76.10     |
> |ODG-Net|97.58	|96.24	|95.23	|90.49	|83.32	|87.90	|82.36	|68.66	|75.80	|62.59	|43.59	|50.22	|83.26	|72.95	|77.29	|
> |MEDIC-cls	|99.03	|95.33	|98.22	|92.06	|83.27	|87.46	|85.62	|69.79	|75.37	|68.40	|41.95	|56.56	|86.28	|72.59	|79.40	|
> |MEDIC-bcls|99.03	|96.04	|97.55	|92.06	|82.68	|87.73	|85.62	|69.15	|76.80	|68.40	|39.60	|55.92	|86.28	|71.87	|79.50	|
> |ours-cls|**99.52**	|**97.30**|99.11|**94.68**|86.10	|92.10	|**89.22**	|**74.31**	|77.76	|**69.49**	|44.34	|55.37	|**88.23**|75.53	|81.09	|
> |ours-bcls|**99.52**	|96.91	|**99.18**	|**94.68**|**88.31**	|**92.28**	|**89.22**	|73.91	|**77.95**	|**69.49**	|**48.09**	|**56.78**	|**88.23**|**76.81**|**81.55**|

---

> > ### Author Response · Authors · 2024-08-12
> > **To Reviewer QWTs**
> >
> > Dear Reviewer QWTs,
> >
> > Thank you for your effort during the review procedure.
> >
> > Since the author-reviewer discussion will be end soon, the authors would like to ask if you have any questions on the rebuttal. We have provided 48 experiments to build the benchmarks on ViT-base and ResNet152 backbone to illustrate the efficacy of our proposed method according to your question.
> >
> > Thank you.
> >
> > Best,
> >
> > The authors.

---

### Official Review · Reviewer_DFZJ · 2024-07-18

**Soundness:** 3
**Presentation:** 3
**Contribution:** 3
**Rating:** 7
**Confidence:** 4

**Summary:**

The paper presents an adaptive domain scheduler with ability to adjust the training order dynamically according to model’s current performance and domain difficulty to address OSDG, short for Open-Set Domain Generalization (OSDG), task where the model is exposed to domain shift and category shift.

**Strengths:**

1.	Effectiveness of addressing limitation of current study. The paper focus on OSDG task and pointed out the problem of existing meta-learning-based OSDG approaches that they didn’t consider influence that the order where domains are presented brings to the process of model generalization. To address this major shortness, the author proposed a novel training strategy named EBiL-HaDS which emphasis the influence of training order and can adjust the order of domain presentation dynamically and automatically.
2.	Abundance of references. The research and experiment is based on abundant and various related work. Summary of existing work especially about domain generalization and open-set recognition is abundant and enough to support his research. With accurate overview and clear awareness of critical shortness of existing study, the paper is revolutionary and meaningful in addressing major problem.
3.	Sufficient and reliable Experiment. The paper conducted several experiments on extensive datasets ,all of which showed dramatically improvement. This showed effectiveness of the proposed method.

**Weaknesses:**

1.	Several mistakes in format such as missing indent before paragraphs and size of pictures, and the formula in between line 161 and line 162.
2.	No open resource code and data. Inconvenience to reproduce and follow the experiment. Also unable to exam the effectiveness of the result.

**Questions:**

1.	Citation in line 91、93 and so on, citation order is not standard.
2.	Line 331~474, The reference format is incorrect, citation of journals needs to be marked with issues, pages and years

**Limitations:**

Experiments show that the method proposed in this paper may still lead to misclassification and biased content. This may cause useless even false predication , causing critical negative influence in popularizing. Additionally, EBiL-HaDS is significantly relied on source domains with unified categories and its using scene is strictly limited, it has only been tested on image classification tasks.

---

> ### Author Rebuttal · Authors · 2024-08-06
>
> Thank you for acknowledging the motivation behind our work and the effort put into the review. We will address the mentioned formatting and citation issues in our final version submission. The source code has been shared in the official comment to the AC, including a sample for result reproduction via an anonymous GitHub link. We hope the provided source code will enhance the contribution of our paper. The included model weight is for the PACS dataset, with the cartoon domain as the target and ResNet18 as the feature extraction backbone. The code will be made publicly available in the final version.

---

> > ### Comment · Reviewer_DFZJ · 2024-08-08
> > **To Author**
> >
> > I can't find the url of the code, and if the code is available, I would like to increase my score.

---

> > > ### Author Response · Authors · 2024-08-08
> > > **Question to AC regarding the url code link and response to the reviewer**
> > >
> > > Dear Reviewer DFZJ, dear AC,
> > >
> > > @Reviewer DFZJ, Thank you for your fast response and we are very glad to hear that you will increase your score. Thank you for the effort of review. We will ask the AC to make the code available to the reviewers.
> > >
> > > @AC, could you please make the source code link we submitted in the official comment to AC available to the reviewers? Thank you.
> > >
> > > According to the rebuttal instruction we have provided the anonymous link of code in the official message to the AC, where the reviewers are not chosen as readers.
> > >
> > > Thank you for your help.
> > >
> > > Best regards,
> > >
> > > The authors.

---

> > > > ### Comment · Reviewer_DFZJ · 2024-08-08
> > > > **To author**
> > > >
> > > > Thank you, I would like to increase my score to 7, and I look forward to the author's future work on DG.

---

### Comment · Area_Chair_MwKT · 2024-08-09
**Anonymous code**

Dear reviewers,

I have checked the anonymity of the source code, which is okay for me. The anonymous GitHub link provided by the authors is available at https://anonymous.4open.science/r/EBiL-HaDS-623E.

Best

---

### Decision · Program_Chairs · 2024-09-25

**Decision:**

Accept (poster)

**Comment:**

While the initial recommendations were divergent, all four reviewers reached a positive consensus (three borderline accept and an accept) after the post-rebuttal discussions. The AC therefore recommends accepting the paper and asks the authors to include their discussions with the reviewers in the final manuscript.